# PRODIGY: AN EXPEDITIOUSLY ADAPTIVE PARAMETER-FREE LEARNER

## ABSTRACT

We consider the problem of estimating the learning rate in adaptive methods, such as Adagrad and Adam. We describe two techniques, Prodigy and Resetting, to provably estimate the distance to the solution $D$, which is needed to set the learning rate optimally. Our techniques are modifications of the D-Adaptation method for learning-rate-free learning. Our methods improve upon the convergence rate of D-Adaptation by a factor of $\mathcal{O}(\sqrt{\log(D/d_0)})$, where $d_0$ is the initial estimate of $D$. We test our methods on 12 common logistic-regression benchmark datasets, VGG11 and ResNet-50 training on CIFAR10, ViT training on Imagenet, LSTM training on IWSLT14, DLRM training on Criteo dataset, VarNet on Knee MRI dataset, as well as RoBERTa and GPT transformer training on BookWiki. Our experimental results show that our approaches consistently outperform D-Adaptation and reach test accuracy values close to that of hand-tuned Adam.

## 1 INTRODUCTION

Optimization is an essential tool in modern machine learning, enabling efficient solutions to large-scale problems that arise in various domains, such as computer vision, natural language processing, and reinforcement learning. One of the key difficulties is the selection of appropriate learning rates, which can significantly impact the convergence speed and the quality of the final solution. Learning-rate tuning has been particularly challenging in applications where there are multiple agents that use their own optimizer. GAN training (Goodfellow et al., 2020), federated learning (Kairouz et al., 2021), and many other settings make this challenge even more pronounced.

In recent years, "parameter-free" adaptive learning rate methods (Orabona & Tommasi, 2017; Cutkosky & Orabona, 2018; Zhang et al., 2022; Carmon & Hinder, 2022; Ivgi et al., 2023) have gained considerable attention due to their ability to automatically adjust learning rates based on the problem structure and data characteristics. Among these, the D-Adaptation method, introduced by (Defazio & Mishchenko, 2023), has emerged as a promising practical approach for learning-rate-free optimization.

D-Adaptation works by maintaining a lower bound on the initial distance to solution $D = \|x_0 - x_*\|$, for any $x_*$ in the solution set of the following problem:

$$\min_{x \in \mathbb{R}^p} f(x).$$

In practice, the lower bound estimated by D-Adaptation increases rapidly during the course of optimization, plateauing to a value close to the true $D$. This $D$ quantity is the key unknown constant needed to set the learning rate for non-smooth optimization methods, forming the numerator of the step size:

$$\gamma_{k+1} = \frac{D}{\sqrt{\sum_{i=0}^{k} \|g_i\|^2}}, \qquad \text{where } D = \|x_0 - x_*\|,$$

and the denominator is based on the Adagrad step size Duchi et al. (2011); Streeter & McMahan (2010); Ward et al. (2019). The Gradient Descent form of D-Adaptation simply plugs in the current lower bound at each step in place of $D$. This simple approach can be

**Algorithm 1** Prodigy (GD version)

1: **Input:** $d_0 > 0$, $x_0$, $G \geq 0$
2: **for** $k = 0$ **to** $n$ **do**
3:   $g_k \in \partial f(x_k)$
4:   Choose weight $\lambda_k$ (default: $\lambda_k = 1$)
5:   $\eta_k = \dfrac{d_k^2 \lambda_k}{\sqrt{d_k^2 G^2 + \sum_{i=0}^k d_i^2 \lambda_i^2 \|g_i\|^2}}$
6:   $x_{k+1} = x_k - \eta_k g_k$
7:   $\hat{d}_{k+1} = \dfrac{\sum_{i=0}^k \eta_i \langle g_i, x_0 - x_i \rangle}{\|x_{k+1} - x_0\|}$
8:   $d_{k+1} = \max(d_k, \hat{d}_{k+1})$
9: **end for**
10: Return $\hat{x}_n = \frac{1}{n+1} \sum_{k=0}^n \eta_k x_k$

**Algorithm 2** Prodigy (Dual Averaging version)

1: **Input:** $d_0 > 0$, $x_0$, $G \geq 0$; $s_0 = 0 \in \mathbb{R}^p$
2: **for** $k = 0$ **to** $n$ **do**
3:   $g_k \in \partial f(x_k)$
4:   $\lambda_k = d_k^2$
5:   $s_{k+1} = s_k + \lambda_k g_k$
6:   $\hat{d}_{k+1} = \dfrac{\sum_{i=0}^k \lambda_i \langle g_i, x_0 - x_i \rangle}{\|s_{k+1}\|}$
7:   $d_{k+1} = \max(d_k, \hat{d}_{k+1})$
8:   $\gamma_{k+1} = \dfrac{1}{\sqrt{\lambda_{k+1} G^2 + \sum_{i=0}^k \lambda_i \|g_i\|^2}}$
9:   $x_{k+1} = x_k - \gamma_{k+1} s_{k+1}$
10: **end for**
11: Return $\bar{x}_n = \frac{1}{n+1} \sum_{k=0}^n \lambda_k x_k$

applied to estimate the step size in Adam (Kingma & Ba, 2015), which yields state-of-the-art performance across a wide-range of deep learning problems. Defazio & Mishchenko (2023) also show that asymptotically, D-Adaptation is as fast as specifying the step size using the true $D$ (up to small constant factors).

In this paper, we present two novel modifications to the D-Adaptation method that enhance its worst-case non-asymptotic convergence rate. By refining the algorithm's adaptive learning rate scheme, we achieve improved performance in terms of convergence speed and solution quality. To validate our proposed modifications, we establish a lower bound for any method that adapts to the distance-to-solution constant $D$. We show that our improved methods are worst-case optimal up to constant factors among methods with exponentially bounded iterate growth. We then conduct extensive experiments that consistently demonstrate that the improved D-Adaptation methods adapt the learning rate much faster than the standard D-Adaptation, leading to enhanced convergence rates and better optimization outcomes.

## 2 PRODIGY APPROACH

To understand how we can improve upon D-Adaptation, let us take a closer look at some nuggets in its analysis. In D-adapted Dual Averaging, the gradient at iteration $k$ is scaled with weight $\lambda_k$. This leads to the error term

$$\text{D-adaptation error} = \sum_{k=0}^n \lambda_k^2 \gamma_k \|g_k\|^2.$$

The theory then proceeds to upper bound this sum using the largest of all $\lambda_k$'s by using the upper bound $\lambda_k \leq \lambda_n$. This, however, is quite pessimistic since then $\lambda_k$ is set to be $\lambda_k = d_k$, so $\lambda_n$ can be as large as $D$ and $\lambda_k$ can be as small as $d_0$. Therefore, replacing $\lambda_k^2$ with $\lambda_n^2$ can introduce a multiplicative error of $\frac{D^2}{d_0^2}$ in this term.

We take a different approach and instead handle the error term using modified Adagrad-like step sizes. In the Adagrad theory, the error term does not have any $\lambda_k^2$ factors, which is exactly why Adagrad places $\sqrt{\sum_{i=0}^k \|g_i\|^2}$ in the step-size denominator. The required modification is then obvious: since the error terms are now $d_i^2 \|g_i\|^2$ instead of $\|g_i\|^2$, the new adaptive step size should be $\gamma_{k+1} = \frac{1}{\sqrt{\sum_{i=0}^k d_i^2 \|g_i\|^2}}$ for the Dual Averaging algorithm and $\eta_k = \frac{d_k^2}{\sqrt{\sum_{i=0}^k d_i^2 \|g_i\|^2}}$ for the Gradient Descent algorithm. This way, we can still control the error term of D-Adaptation but the obtained step size is provably larger since $d_k$ is non-decreasing. Having larger step sizes while preserving the main error term is the key reason why the new algorithms converge, as we show below, with a faster rate.

Notice, however, that the methods might still be slow because the denominator in the step size might grow too large over time. To remedy this, we introduce a modification for the Gradient Descent step size by placing an extra weight $\lambda_k$ next to the gradients:

$$\eta_k = \frac{d_k^2 \lambda_k}{\sqrt{\sum_{i=0}^{k} d_i^2 \lambda_i^2 \|g_i\|^2}}.$$

In fact, the modified step size might even increase between iterations, whereas the Adagrad step size always decreases. We will show that as long as $\lambda_k$ does not grow too quickly, the worst-case convergence rate is almost the same.

To have non-asymptotic theory, we also introduce in our algorithms an extra term $G^2$ in the denominator which upper bound the gradient norm. We define it formally in the assumption below.

**Assumption 1.** *We assume that the objective $f$ is $G$-Lipschitz, which implies that its gradients are bounded by $G$: for any $x \in \mathbb{R}^p$ and $g \in \partial f(x)$, it holds $\|g\| \leq G$.*

Algorithm 1 and Algorithm 2 give Gradient Descent and the Dual Averaging variants of our new method. In contrast to Adagrad, they estimate the *product* of $D$ and $G$ in the denominator, so we call the proposed technique *Prodigy*. We give the following convergence result for Algorithm 1:

**Theorem 1.** *Assume $f$ is convex and $G$-Lipschitz. Given any weights $1 \leq \lambda_0 \leq \cdots \leq \lambda_n$, the functional gap of the average iterate of Algorithm 1 converges as*

$$f(\hat{x}_n) - f_* \leq \sqrt{2\lambda_n} D G \frac{2 d_{n+1} + d_{n+1} \log(1 + \sum_{k=0}^{n} \lambda_k^2)}{\sqrt{\sum_{k=0}^{n} \lambda_k d_k^2}}, \tag{1}$$

*where $\hat{x}_n = \frac{1}{n+1} \sum_{k=0}^{n} \eta_k x_k$ is the weighted average iterate.*

Notice that we have the freedom to choose any non-decreasing sequence $\lambda_k$ as long as the right-hand side is decreasing, e.g., by setting $\lambda_k = k^p$ with $p \geq 0$. This allows us to put much more weight on the recent gradients and get more reasonable step sizes. While it is not guaranteed to be better in theory, it is usually quite important to do so in practice.

In contrast to the bound in Defazio & Mishchenko (2023), we bound $\frac{d_{t+1}}{\sqrt{\sum_{k=0}^{t} d_k^2}}$ instead of $\frac{d_{t+1}}{\sum_{k=0}^{t} d_k}$. This is the reason why the overall guarantee improves by a factor of $\sqrt{\log_2(D/d_0)}$. For instance, if we set $\lambda_k = 1$ for all $k$ and substitute the bound from Lemma 1, we get the convergence rate

$$f(\hat{x}_t) - f_* = \mathcal{O}\left( \frac{G D \log(n+1) \sqrt{\log_{2+}(D/d_0)}}{\sqrt{n}} \right).$$

where $t \leq n$ is chosen as the argmin from Lemma 1. Even though our theory does not guarantee that it is beneficial to use increasing weights $\lambda_k$, our result is, to the best of our knowledge, new for Adagrad-like methods. It allows for a wide range of choices in $\lambda_k$. For example, if we set $\lambda_k = \beta_2^{-k^p}$ with $\beta_2 < 1$ and $p < 1/3$, then the method is still guaranteed to converge at the rate of $\mathcal{O}\left( \frac{1}{n^{(1-3p)/2}} \right)$. This is of particular interest when we study Adam-like methods, see Appendix B for an additional discussion.

The logarithmic term $\log(n+1)$ is, however, not necessary and only arises due to the use of Gradient Descent update. The Dual Averaging update, provides a tighter guarantee as given in the next theorem.

**Theorem 2.** *Let $f$ be a convex and $G$-Lipschitz function. For Algorithm 2, it holds that:*

$$f(\overline{x}_t) - f_* \leq \frac{4 G D}{\sqrt{n}} \sqrt{\log_{2+}\left(\frac{D}{d_0}\right)},$$

*where $t = \arg\min_{k \leq n} \frac{d_{k+1}}{\sqrt{\sum_{i=0}^{k} d_i^2}}$ and $\log_{2+}(\cdot) = 1 + \log_2(\cdot)$.*

Comparing this with the previous rate, the only difference is the removal of a multiplicative $\log(n+1)$ factor. This improvement, however, is mostly theoretical as Gradient Descent typically performs better in practice than Dual Averaging. We also note that we do not have a convergence result for Algorithm 2 with weights other than $\lambda_k = d_k^2$. This is due to the fact that the DA analysis requires the step size to be monotonically decreasing, so we cannot place an extra weighting factor in the numerator of $\gamma_{k+1}$.

## 3 Resetting Approach

---

**Algorithm 3** D-Adaptation with Resetting

---

1: **Input:** $d_0 > 0$, $x_0$, $G$
2: $s_0 = 0$, $r = 0$, $k = 0$, $x_{r,0} = x_0$, $s_{0,0} = 0$
3: **for** $j = 0$ **to** $n$ **do**
4:      $g_{r,k} = g_j \in \partial f(x_j)$
5:      $s_{r,k+1} = s_{r,k} + g_{r,k}$
6:      $\gamma_{r,k+1} = \dfrac{d_r}{\sqrt{G^2 + \sum_{i=0}^{k} \|g_{r,i}\|^2}}$
7:      $\hat{d}_{r,k+1} = \dfrac{\sum_{i=0}^{k} \gamma_i \langle g_{r,i},\, s_{r,i} \rangle}{\|s_{r,k+1}\|}$
8:      **if** $\hat{d}_{r,k+1} > 2d_r$ **then** start a new epoch with $g_{r,k}$ as the first gradient of the epoch
9:          $d_{r+1} = \hat{d}_{r,k+1}$, $x_{0,r+1} = x_0$, $g_{r+1,0} = g_{r,k}$, $s_{r+1,1} = g_{r+1,0}$
10:          $\gamma_{r+1,1} = \dfrac{d_{r+1}}{\sqrt{G^2 + \|g_{r+1,0}\|^2}}$
11:          $x_{j+1} = x_{r+1,1} = x_{r+1,0} - \gamma_{r+1,1} s_{r,1}$
12:          $k = 0$, $r = r + 1$
13:      **else**
14:          $x_{j+1} = x_{r,k+1} = x_{r,0} - \gamma_{r,k+1} s_{r,k+1}$
15:      **end if**
16:      $k = k + 1$
17: **end for**
18: Return $\bar{x}_n = \frac{1}{n+1} \sum_{j=0}^{n} x_j$

---

Algorithm 3 describes a variant of D-Adaptation where the Dual Averaging process is reset whenever the current $d_k$ estimate increases by more than a factor of 2. We call the interval between resetting events an *epoch*. This resetting process has a number of other effects:

- The step-size sequence $\gamma$ is also reset, resulting in larger steps right after the reset.
- The convergence of the method is proven with respect to an unweighted average of the iterates, rather than a weighted average.
- Since the quantities tracked to compute $\hat{d}$ are also reset, the value of $\hat{d}$ often will increase more rapidly than it can when using the standard D-Adaptation estimate.

This resetting variant has the advantage of being significantly simpler to analyze in the non-asymptotic case than standard D-Adaptation or Prodigy. This makes it well suited to be used as a basis for extensions and modifications of D-Adaptation.

**Theorem 3.** *Under the assumption of convex and G-Lipschitz $f$, we have for Algorithm 3:*

$$f(\bar{x}_n) - f_* \leq \frac{6DG\sqrt{\log_{2+}(D/d_0)}}{\sqrt{n+1}}.$$

This is the same rate as we established for the Dual Averaging variant of Prodigy, but we return the average of all iterates, rather than an average computed up to some point $t \leq n$, a significant simplification. However, due to the resetting operation, this method is not expected to work as well as Prodigy in practice.

## 4 Complexity Lower Bound

A lower complexity bound can be established for the Lipschitz-Convex complexity class via a simple 1-dimensional resisting oracle. The bound depends on the "scale" of the initial step of the algorithm, which is the size of the initial step from $x_0$ to $x_1$. This initial step

is $g_0 \cdot d_0 / \sqrt{G^2 + \|g_0\|^2}$ for D-Adaptation, and can be though of as an algorithm-agnostic measure of $d_0$.

Our lower bound allows the resisting oracle to choose a constant $D$ after seeing the iterates, which is a much stronger oracle then required for establishing a lower bound. Ideally, a lower bound could be established where the constant $D$ is fixed but unknown to the algorithm, and the actual distance to solution $\|x_0 - x_*\| \le D$ given by the oracle is allowed to depend on the iterate sequence.

The primary consequence of this difference is that our construction only tells us that hard problems exist for $n$ small relative to $D/d_0$, of the scale $n < \log\log(D/d_0)$. It remains an open problem to show a lower bound for larger $n$.

**Theorem 4.** *Consider any Algorithm for minimizing a convex $G$-Lipschitz function starting from $x_0$ at the origin, which has no knowledge of problem constants. At each iteration $k$, the algorithm may query the gradient at a single point $x_k$. Then for any sequence of $x_{1,\dots,}\,x_n$, there exists a convex Lipschitz problem $f$ and constant $D \ge \|x_0 - x_*\|$ for all minimizers $x_*$ of $f$ such that:*

$$\min_{k \le n} f(x_k) - f_* \ge \frac{DG\sqrt{\log_2 \log_2(D/x_1)}}{2\sqrt{n+1}}.$$

Carmon & Hinder (2022) (Sec 3.3) give a method with a matching upper bound in the $G$-Lipschitz, non-stochastic case. It's currently an open problem to find a method that avoids any additional multiplicative log factors asymptotically, while at the same time giving a $\sqrt{\log(\log(\cdot))}$ any-time rate.

Lower complexity bounds for the average regret in the more general online learning setting also apply here. They are of the form (Zhang et al., 2022):

$$\frac{1}{n}\sum_{k=0}^{n} \langle g_k, x_k - x_* \rangle = \Omega\left(\frac{DG\sqrt{\log_2(D/\epsilon)} + \epsilon}{\sqrt{n+1}}\right).$$

where $\epsilon$ is a "user-specified constant" playing a similar role to $x_1$. Bounds on the average regret directly bound function value sub-optimality as

$$f(\bar{x}) - f_* \le \frac{1}{n+1}\sum_{k=0}^{n}[f(x_k) - f_*] \le \frac{1}{n+1}\sum_{k=0}^{n}\langle g_k, x_k - x_* \rangle,$$

where $\bar{x} = \frac{1}{n+1}\sum_{k=0}^{n} x_k$.

## 4.1 Exponentially Bounded Algorithms

The lower bound construction above applies to algorithms generating sequences of iterates growing arbitrary fast. We can obtain an interesting class of algorithms, which contains our two D-Adaptation variants, by restricting the rate of growth.

**Definition 1.** *An optimization algorithm is exponentially bounded if there exists a constant $d_0$, so that for any sequence of $G$-bounded gradients it returns a sequence of iterates such that for all $k$:*

$$\|x_k - x_0\| \le 2^k d_0.$$

**Theorem 5.** *D-Adaptation, DoG, Prodigy and D-Adaptation with resetting are exponentially bounded.*

Our new D-Adaptation variants are optimal among exponentially bounded algorithms for this complexity class:

**Theorem 6.** *Consider any exponentially bounded algorithm for minimizing a convex $G$-Lipschitz function starting from $x_0$, which has no knowledge of problem constants $G$ and $D$. There exists a fixed gradient oracle such that any sequence of $x_{1,\dots,}\,x_n$, there exists a convex Lipschitz problem $f$ with $G = 1$ and $\|x_0 - x_*\| \le D$ for all minimizing points $x_*$, consistent with the gradient oracle such that:*

$$\min_{k \le n} f(x_k) - f_* \ge \frac{DG\sqrt{\log_2(D/x_1)}}{2\sqrt{n+1}}.$$

---

**Algorithm 4** Prodigy (Adam version)

---

1: **Input:** $d_0 > 0$ (default $10^{-6}$), $x_0$, $\beta_1$ (default 0.9), $\beta_2$ (default 0.999), $\epsilon$ (default $10^{-8}$), $\gamma_k$ (default 1 with cosine annealing)
2: $r_0 = 0$, $s_0 = 0$, $m_0 = 0$, $v_0 = 0$
3: **for** $k = 0$ **to** $n$ **do**
4:      $g_k \in \partial f(x_k)$
5:      $m_{k+1} = \beta_1 m_k + (1 - \beta_1) d_k g_k$
6:      $v_{k+1} = \beta_2 v_k + (1 - \beta_2) d_k^2 g_k^2$
7:      $r_{k+1} = \sqrt{\beta_2} r_k + (1 - \sqrt{\beta_2}) \gamma_k d_k^2 \langle g_k, x_0 - x_k \rangle$
8:      $s_{k+1} = \sqrt{\beta_2} s_k + (1 - \sqrt{\beta_2}) \gamma_k d_k^2 g_k$
9:      $\hat{d}_{k+1} = \dfrac{r_{k+1}}{\|s_{k+1}\|_1}$
10:     $d_{k+1} = \max(d_k, \hat{d}_{k+1})$
11:     $x_{k+1} = x_k - \gamma_k d_k m_{k+1} / (\sqrt{v_{k+1}} + d_k \epsilon)$
12: **end for**

---

Using the simple construction from Theorem 6, we show in Appendix E that the class of exponentially bounded methods (potentially with an exponent other than 2) covers all Gradient Descent approaches that use an estimate of $d_k \leq cD$ for some constant c, and use a step size $\gamma_k \leq d_k/G$ without line-search or other additional queries. So the only way to achieve a $\log\log$ dependence on $d_0$ is by using a method that performs some queries that overshoot the standard $D/G$ step size by more than a fixed constant factor. Although using larger step sizes is not problematic for Lipschitz functions, it comes with the risk of causing training divergence when applied to functions whose gradients are only locally bounded by $G$, which is common in machine learning settings.

## 5 Related Work

In this section, we review the major classes of techniques for optimizing convex Lipschitz functions with some level of problem parameter independence.

The Polyak step size Polyak (1987) trades the knowledge of $D$ for $f_*$, achieving optimal convergence rate without additional log factors. Stable convergence requires accurate $f_*$ estimates. A restarting scheme converges within a multiplicative log factor of the optimal rate Hazan & Kakade (2019). There has been substantial recent research on modifications of the Polyak step size to make it better suited to machine learning tasks (Loizou et al., 2021; Gower et al., 2021; Orvieto et al., 2022) but as of yet they have not seen widespread adoption.

Coin-betting Orabona & Tommasi (2017); McMahan & Orabona (2014); Cutkosky & Orabona (2018); Zhang et al. (2022); Orabona & Pál (2021) is a family of approaches from the online learning setting which are also applicable for convex non-smooth optimization. They work by establishing a relationship by duality between regret minimization and wealth-maximization. Existing approaches for wealth-maximization can then be mapped to algorithms for regret minimization. Coin-betting approaches give convergence rates for an equal-weighted average of the iterates of the form:

$$f(\bar{x}_n) - f_* = \mathcal{O}\left(\frac{DG\sqrt{\log(1 + D/d_0)}}{\sqrt{n+1}}\right).$$

Standard D-Adaptation obtains asymptotic rates without the log factor, but was otherwise (theoretically) slower in finite time, as it had a $\log(\cdot)$ rather than a $\sqrt{\log(\cdot)}$ dependence on $D/d_0$:

$$f(\hat{x}_n) - f_* \leq \frac{16 \log_{2+}(d_{n+1}/d_0)}{n+1} D \sqrt{\sum_{k=0}^{n} \|g_k\|^2} \leq \frac{16DG \log_{2+}(D/d_0)}{\sqrt{n+1}}.$$

Our two new variants close this gap, giving the same sqrt-log dependence as coin betting.

The DoG method (Ivgi et al., 2023), based on the bisection approach of Carmon & Hinder (2022), is the only other approach that we are aware of that estimates $D$ in an online fashion.

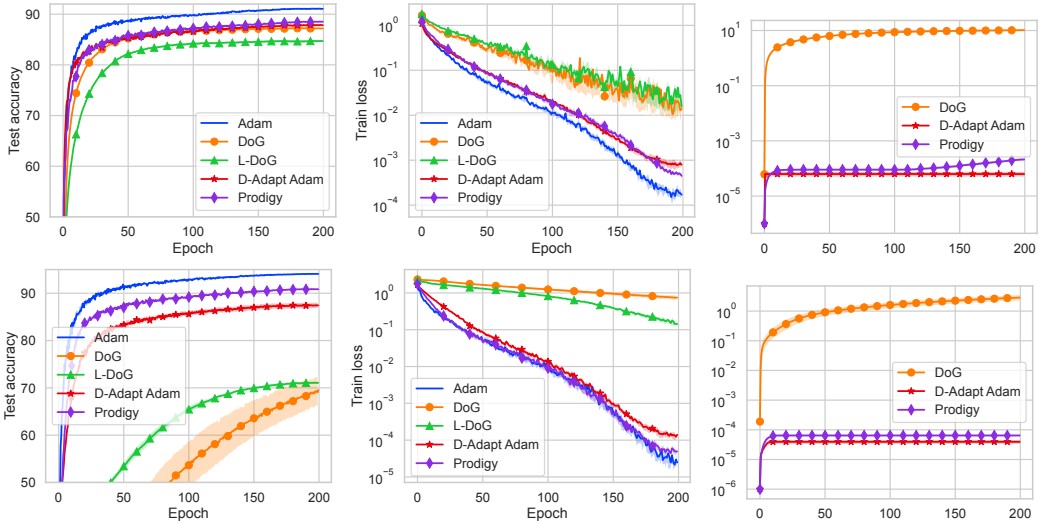

Figure 1: VGG11 and ResNet-50 training on CIFAR10. Left: test accuracy (%), middle: train loss, right: step sizes. "Prodigy" is used as given in Algorithm 4. As expected, Prodigy estimates a larger step size than D-Adaptation, which helps it reach test accuracy closer to the one of Adam.

DoG estimates $D$ by $\bar{r}_k$: $\bar{r}_k = \max_{i \leq k} \|x_i - x_0\|$. Ivgi et al. (2023) use this quantity as a plug-in estimate for the numerator of the step size, similar to D-Adaptation's approach. This approach can divergence in theory, but with additional modifications to the step size, the "tamed" T-DoG method is shown to converge. It has a $\log_+(D/d_0)$ dependence on the initial sub-optimally of the D estimate, so our approach improves on this dependence by a $\sqrt{\log_+(D/d_o)}$ factor.

Malitsky & Mishchenko (2020) proposed AdGD, a method for convex optimization that does not require any hyperparameters and has a rate that is at least as good as that of the optimally tuned Gradient Descent. However, AdGD requires the objective to be locally smooth, which hinders its use in many practical problems. Latafat et al. (2023) partially addressed this gap by proposing a proximal extension, but the case of non-smooth differentiable functions has remained unstudied.

## 6 EXPERIMENTS

We test our methods on convex logistic regression as well as deep learning problems. In all deep learning experiments, the Prodigy method is used as presented in Algorithm 4, whose derivation is explained in Appendix B. We provide detailed descriptions of our experiments in Appendix A and discuss the results here.

**Logistic regression.** We performed 1,000 steps for each dataset, using a randomized $x_0$ and plot the results of 10 seeds. We ran both DA and SGD variants of each method. Each plot shows the accuracy of the average iterate for each method. Due to limited space, we provide the results in Appendix A.1. In short, our proposed algorithms greatly out-perform regular D-Adaptation. Our weighted SGD variant of D-Adaptation is faster consistently across each dataset. Additionally, it adapts faster than the DoG method (Ivgi et al., 2023) on 10 of the 12 problems.

**CIFAR10.** For neural network experiments , we consider training on CIFAR10 (Krizhevsky, 2009) with batch size 256, where D-Adapted Adam has a gap of a few percent compared to the standard Adam. We use cosine annealing with initial step size 1 for all Adam-based methods and initial step size $10^{-3}$ for Adam itself. The considered networks are VGG11

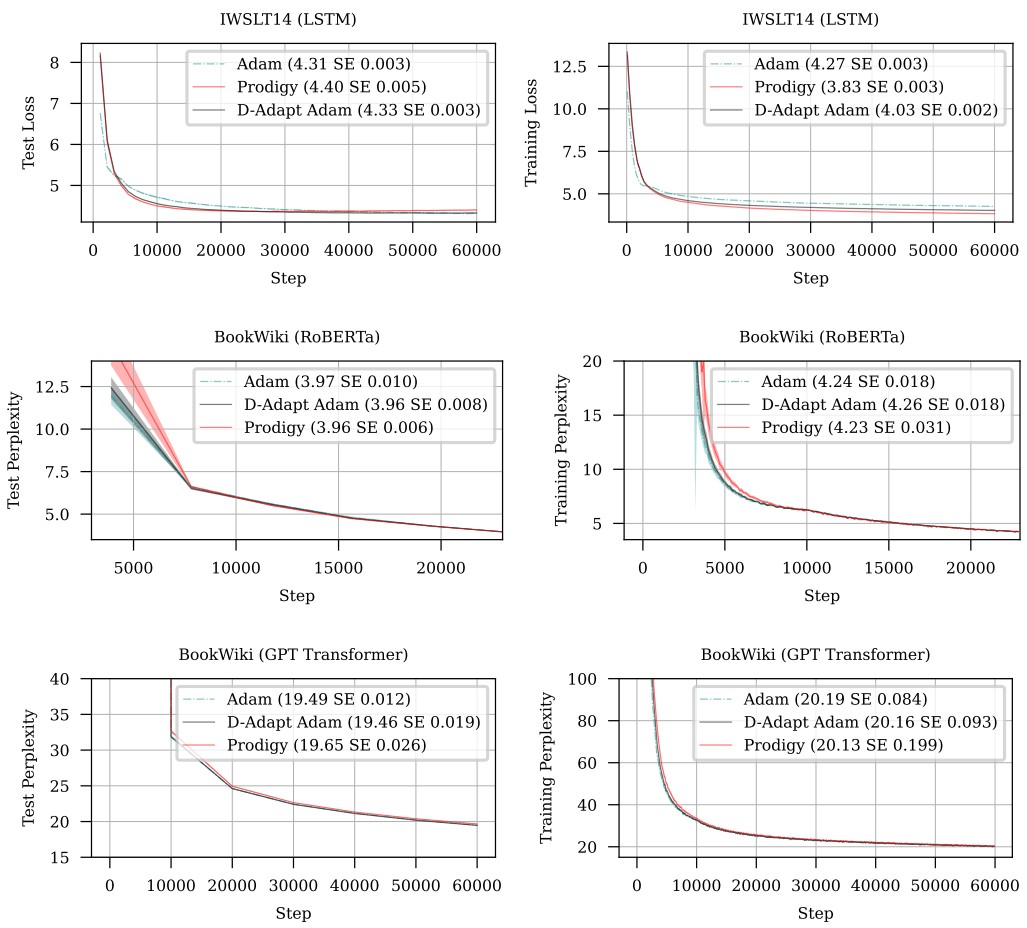

Figure 2: Adam-family experiments.

(Simonyan & Zisserman, 2014) and ResNet-50 (He et al., 2016)[1]. To simplify the experiment, we do not use weight decay, so both networks slightly overfit and do not reach high test accuracy values. All methods were run using same 8 random seeds.

We show the results in Figure 1. As we can see, this gap is closed by Prodigy, which is achieved by the larger estimates of the step size.

**nanoGPT transformer.** We also train a 6-layer transformer network from nanoGPT[2] on the Shakespeare dataset, with the results provided and discussed in Appendix A.

## 6.1 LARGE-SCALE ADAM EXPERIMENTS

To validate the performance on large-scale practical applications directly against D-Adaptation, we ran the subset of the experiments from Defazio & Mishchenko (2023) that use the Adam optimizer. Methods without coordinate adaptivity are known to underperform on these problems and so we exclude SGD and DoG from these comparisons.

**LSTM, RoBERTa, GPT, DLRM, VarNet.** On the smallest problem of LSTM training, Prodigy appears to converge significantly faster in training loss and slightly overfits in test

---

[1]VGG11 and ResNet-50 implementation along with the data loaders were taken from `https://github.com/kuangliu/pytorch-cifar`

[2]`https://github.com/karpathy/nanoGPT`

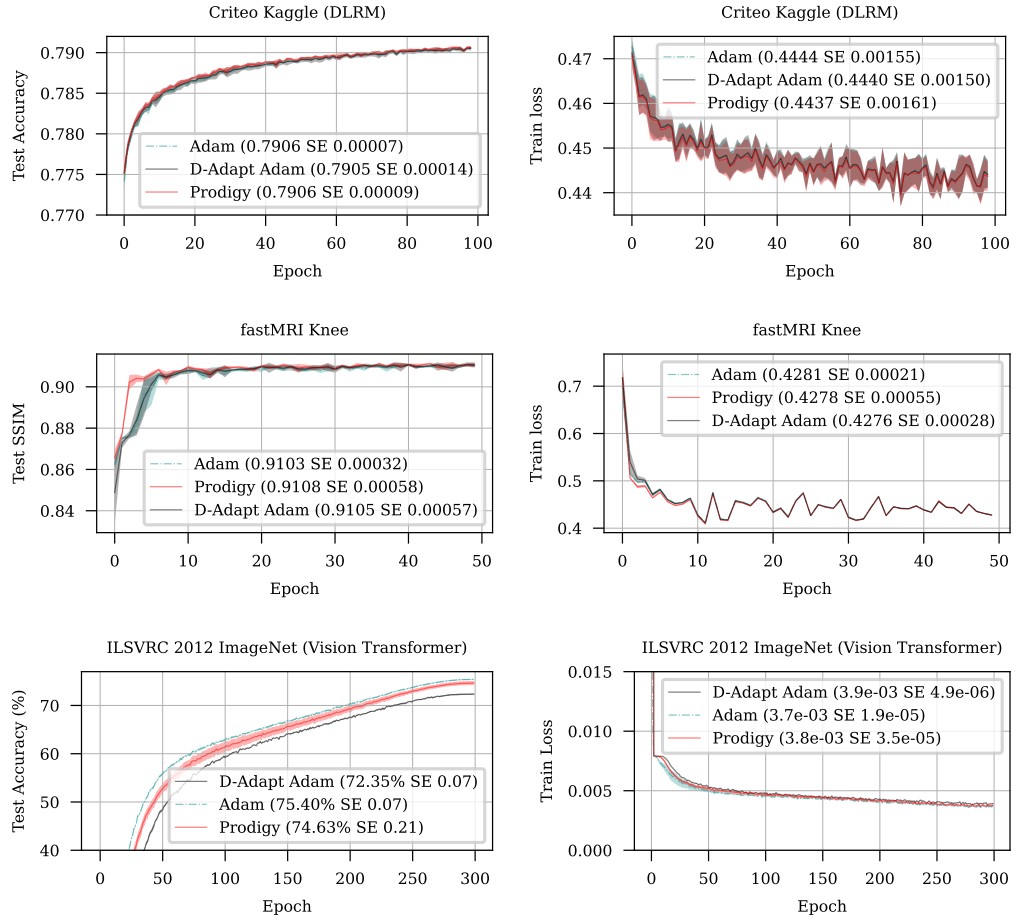

Figure 3: Adam-family experiments.

loss compared to the baselines. For RoBERTa (Liu et al., 2019) and GPT (Radford et al., 2019) training on BookWiki, Prodigy matches the performance of the baseline with only negligible differences. For the application problems, DLRM (Naumov et al., 2019) on the Criteo Kaggle Display Advertising dataset, and fastMRI VarNet (Zbontar et al., 2018), Prodigy again closely matches the baselines.

**ViT training.** Defazio & Mishchenko (2023) present a negative result for training vision transformer (Dosovitskiy et al., 2021), where D-Adaptation significantly underperforms tuned Adam. We were able to reproduce this gap across a wide range of weight-decay values, although this problem has high run-to-run variance of 1-2% of test accuracy, which makes comparison difficult. We can see that Prodigy almost closes the gap between tuned Adam and D-Adaptation, giving a test accuracy of 74.63% compared to 75.4% for Adam, and more than 2% higher than D-Adaptation. See Figure 6.1 for the results.

## 7 CONCLUSION

We have presented two new methods for learning rate adaptation that improve upon the adaptation rate of the state-of-the-art D-Adaptation method. Prodigy, a form of weighted D-Adaptation, was shown to adapt faster than other known methods across a range of experiments. The second method, D-Adaptation with resetting, is shown to achieve the same theoretical rate as Prodigy with a much simpler theory than Prodigy or even D-Adaptation.

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

# A   EXTRA EXPERIMENTAL DETAILS AND DISCUSSION

## A.1   LOGISTIC REGRESSION EXPERIMENTS

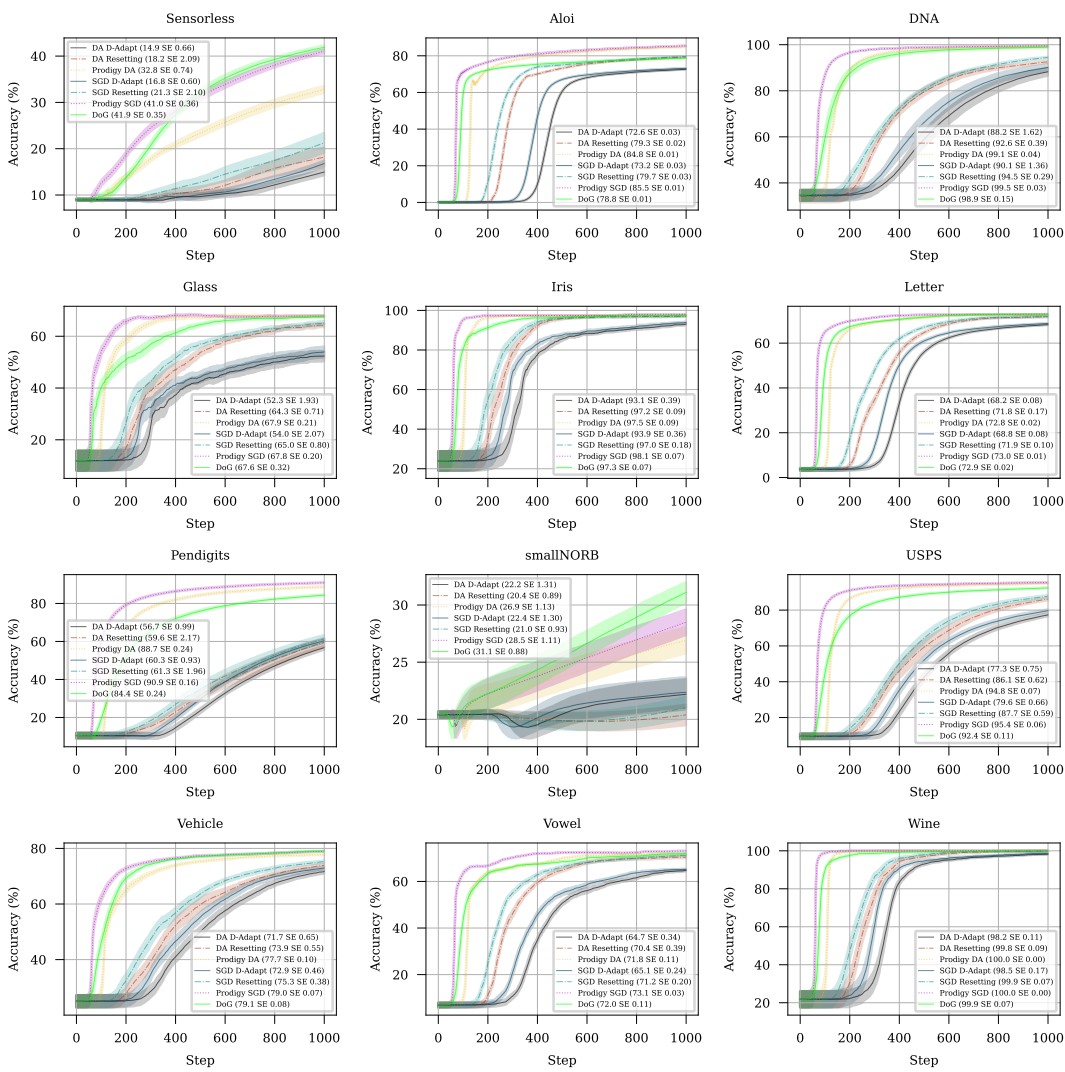

Figure 4:   Convex multiclass classification problems. Error bars show a range of 1 standard error above and below the mean of the 10 seeds.

**Logistic regression.**   For the convex setting, we ran a set of classification experiments. For each dataset, we used the multi-margin loss (Weston & Watkins, 1999), a multi-class generalization of the hinge loss. This non-smooth loss results in bounded gradients, which is required by our theory. We perform full-batch rather that stochastic optimization, for two reasons. Firstly, it matches the assumptions of our theory. Secondly, fast learning rate adaptation is more crucial for full-batch optimization than stochastic optimization as fewer total steps will be performed.

## A.2   NEURAL NETWORKS

**CIFAR10.**   For DoG and L-DoG, we compute the polynomial-averaging iterate and then report the best of the average and the last iterate. We average with $\gamma = 8$, see (Ivgi et al., 2023) for the details. While DoG produces larger step size estimate than Prodigy (see the

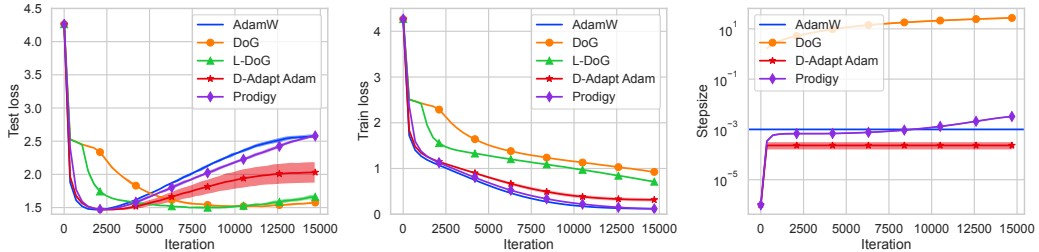

Figure 5: The test (left) and train (middle) loss curves as well as the estimated stepsize (right) when training a 6-layer nanoGPT transformer on the Shakespeare dataset.

right column in Figure 1, this is counterbalanced by the larger denominator in DoG. We also tried to modify DoG to use Adam-like step sizes but our heuristic modification diverged on this problem. We also observed that among DoG and its layer-wise version, L-DoG, there is no clear winner as the former performed better on VGG11 and the latter was better when training ResNet-50.

**nanoGPT.** For all methods, we use batch size 256, clip the gradients to have norm not exceeding 1 and use float16 numbers. We use AdamW with hyperparameters given in the repository, i.e., $\beta_2 = 0.99$, weight decay 0.1, stepsize $10^{-3}$, cosine annealing with warmup over 100 steps. The same weight decay value and cosine annealing is used for Prodigy and D-Adapted Adam, except that the latter two methods use stepsize 1. We accumulate minibatches of size 12 into a batch of size 480. We tuned the weight decay for DoG and L-DoG and found the value $10^{-4}$ to work well for this problem. We ran each method with 8 random seeds and report the average as well as one-standard-deviation confidence intervals.

See Figure 5 for the results. In terms of the test loss, all methods are roughly equivalent except that DoG and L-DoG were slower to reach the best value of roughly 1.5. For the train loss, Prodigy was on par with tuned AdamW and slightly better than D-Adapted Adam. Surprisingly, the estimated step size in Prodigy was very consistent across the 8 random seeds.

**ViT training.** We tested several values of weight decay on top of the 0.1 value reported by Defazio & Mishchenko (2023). Using weight decay 0.05 instead of 0.1 significantly improved performance of each variant, and so we present results for both the baselines and Prodigy at that value.

## B  DERIVING ADAM-LIKE STEP SIZES

To derive an Adam-like method, which should use exponential moving average for the step size, we want to approximate Adam's update of the exponential moving average of squared gradients: $v_{k+1} = \beta_2 v_k + (1 - \beta_2)g_k^2 = (1 - \beta_2)\sum_{i=0}^{k} \beta_2^{k-i} g_i^2$, where $g_k^2$ is the coordinate-wise square of the gradient $g_k$. We can achieve this using exponential weights, $\lambda_k = \beta_2^{-k/2}$, which after substituting the definition of $\eta_k$ give us the following identity:

$$\frac{d_k^4}{\eta_k^2} = \frac{d_k^2}{\lambda_k^2}G^2 + \sum_{i=0}^{k} d_i^2 \frac{\lambda_i^2}{\lambda_k^2}\|g_i\|^2 = \frac{d_k^2}{\lambda_k^2}G^2 + d_k^2\|g_k\|^2 + \sum_{i=0}^{k-1} \beta_2^{k-i}d_i^2\|g_i\|^2.$$

This can be seen as computing an exponential moving average of $d_k g_k$ rather than $g_k$ itself. This is our first observation. In addition, in Appendix C.5, we provide a coordinate-wise version of Algorithm 2 and study its convergence properties. Based on the theory presented there, the denominator for $\hat{d}_{k+1}$ should use the $\ell_1$ norm of the weighted gradient sum. Thus, combining this insight with the design of Algorithm 1, we obtain the following expression for

the Adam estimate of $D$:

$$\hat{d}_{k+1} = \frac{\sum_{i=0}^k \lambda_i d_i^2 \langle g_i, x_0 - x_i \rangle}{\| \sum_{i=0}^k \lambda_i d_i^2 g_i \|_1} = \frac{\sum_{i=0}^k \beta_2^{(k-i)/2} d_i^2 \langle g_i, x_0 - x_i \rangle}{\| \sum_{i=0}^k \beta_2^{(k-i)/2} d_i^2 g_i \|_1}.$$

The update uses exponential moving average as well, although it is more conservative as it uses $\sqrt{\beta_2}$ instead of $\beta_2$. Note that there is an extra of $(1 - \beta_2)$ in the update for $v_k$, which can be optionally compensated for by using the bias correction discussed by Kingma & Ba (2015). These update rules are summarized in Algorithm 4. This is the main algorithm that we study numerically in the next section.

## C   ANALYSIS OF PRODIGY

As a reminder, we use the notation $\log_{2+}(a) = 1 + \log_2(a)$ to denote the logarithm that is lower bounded by 1 for any $a \geq 1$.

### C.1   USEFUL PROPOSITIONS

**Proposition 1** (Lemma A.2 in Levy et al. (2018)). *For any sequence of nonnegative real numbers $a_0, \ldots, a_n$*

$$\sqrt{\sum_{k=0}^n a_i} \leq \sum_{k=0}^n \frac{a_k}{\sqrt{\sum_{i=0}^k a_i}} \leq 2\sqrt{\sum_{k=0}^n a_i}. \tag{2}$$

*Proof.* For completeness, we prove both statements here. Notice that for any $\alpha \in [0,1]$, it holds $1 - \sqrt{1-\alpha} \leq \alpha \leq 2(1 - \sqrt{1-\alpha})$. Substituting $\alpha = \frac{a_k}{\sum_{i=0}^k a_i}$ gives

$$1 - \sqrt{1 - \frac{a_k}{\sum_{i=0}^k a_i}} \leq \frac{a_k}{\sum_{i=0}^k a_i} \leq 2\left(1 - \sqrt{1 - \frac{a_k}{\sum_{i=0}^k a_i}}\right).$$

If we multiply all sides by $\sqrt{\sum_{i=0}^k a_i}$, the inequality above becomes

$$\sqrt{\sum_{i=0}^k a_i} - \sqrt{\sum_{i=0}^{k-1} a_i} \leq \frac{a_k}{\sqrt{\sum_{i=0}^k a_i}} \leq 2\left(\sqrt{\sum_{i=0}^k a_i} - \sqrt{\sum_{i=0}^{k-1} a_i}\right).$$

Summing over $k = 0, \ldots, n$, we get the stated bound. $\square$

**Proposition 2.** *For any sequence of nonnegative numbers $a_0, \ldots, a_n$ and $A > 0$, it holds*

$$\sum_{k=0}^n \frac{a_k}{A + \sum_{i=0}^k a_i} \leq \log\left(A + \sum_{k=0}^n a_k\right) - \log(A). \tag{3}$$

*Proof.* If $a_i = 0$ for some $i$, we can simply ignore the corresponding summands, so let us assume that $a_i > 0$ for all $i$. For any $t > 0$ it holds $1/(1+t) \leq \log(1 + 1/t)$. Substituting $t = S_k/a_k$, where $S_k = A + \sum_{i=0}^{k-1} a_i$ for $k > 0$ and $S_0 = A$, we get

$$\frac{1}{1 + \frac{S_k}{a_k}} = \frac{a_k}{a_k + S_k} = \frac{a_k}{A + \sum_{i=0}^k a_i} \leq \log(1 + a_k/S_k) = \log(S_{k+1}) - \log(S_k).$$

Summing this over $k$ from 0 to $n$, we get

$$\sum_{k=0}^n \frac{a_k}{A + \sum_{i=0}^k a_i} \leq \sum_{k=0}^n (\log(S_{k+1}) - \log(S_k)) = \log(S_{n+1}) - \log(S_0)$$

$$= \log\left(A + \sum_{k=0}^n a_k\right) - \log(A).$$

This is exactly what we wanted to prove. $\square$

**Lemma 1.** *Let $d_0 \leq d_1 \leq \cdots \leq d_N$ be positive numbers and assume $N \geq 2\log_{2+}\left(\frac{d_N}{d_0}\right)$, where $\log_{2+}(\cdot) = 1 + \log_2(\cdot)$. Then,*

$$\min_{t<N} \frac{d_{t+1}}{\sqrt{\sum_{k=0}^{t} d_k^2}} \leq \frac{4\sqrt{\log_{2+}\left(\frac{d_N}{d_0}\right)}}{\sqrt{N}}.$$

### C.2 Proof of Lemma 1

*Proof.* Following the proof in Ivgi et al. (2023), we define $K = \left\lceil \log_2\left(\frac{d_N}{d_0}\right) \right\rceil$ and $n = \left\lfloor \frac{N}{K} \right\rfloor$. Consider a partitioning of the sequence $t \leq N$ into half-open intervals $I_k = [nk, n(k+1))$ for $k = 0$ to $K - 1$. We want to show that there is at least one interval such that $d_k$ changes by at most a factor of 2 on that interval. We will use proof by contradiction.

Suppose that for all intervals, $d_{nk} < \frac{1}{2}d_{n(k+1)}$. Then $d_k$ at least doubles in every interval, and so:

$$d_0 < \frac{1}{2}d_n < \frac{1}{4}d_{2n} \cdots < \frac{1}{2^K}d_{nK} < \frac{1}{2^K}d_N,$$

which implies that $d_N/d_0 > 2^K$ and so $K < \log_2(d_N/d_0)$ which contradictions our definition $K = \left\lceil \log_2\left(\frac{d_N}{d_0}\right) \right\rceil$. Therefore, there exists some $\hat{k}$ such that $d_{n\hat{k}} \geq \frac{1}{2}d_{n(\hat{k}+1)}$. We can now proceed with proving the Lemma by considering the summation over interval $I_{\hat{k}}$ only:

$$\min_{t<N} \frac{d_{t+1}}{\sqrt{\sum_{k=0}^{t} d_k^2}} \leq \frac{d_{n(\hat{k}+1)}}{\sqrt{\sum_{k=0}^{n(\hat{k}+1)-1} d_k^2}} \leq \frac{d_{n(\hat{k}+1)}}{\sqrt{\sum_{k=n\hat{k}}^{n(\hat{k}+1)-1} d_k^2}} \leq \frac{d_{n(\hat{k}+1)}}{\sqrt{\sum_{k=n\hat{k}}^{n(\hat{k}+1)-1} d_{n\hat{k}}^2}}$$

$$= \frac{d_{n(\hat{k}+1)}}{\sqrt{nd_{n\hat{k}}^2}} \leq \frac{d_{n(\hat{k}+1)}}{\sqrt{\frac{1}{4}nd_{n(\hat{k}+1)}^2}} = \frac{2}{\sqrt{n}} = \frac{2}{\sqrt{\left\lfloor \frac{N}{K} \right\rfloor}}$$

$$\leq \frac{2}{\sqrt{\frac{N}{K}-1}} \leq \frac{2}{\sqrt{\frac{N}{\log_2(d_N/d_0)+1}-1}} = \frac{2\sqrt{\log_{2+}\left(\frac{d_N}{d_0}\right)}}{\sqrt{N-\log_{2+}\left(\frac{d_N}{d_0}\right)}}$$

$$\overset{N \geq 2\log_{2+}\left(\frac{d_N}{d_0}\right)}{\leq} \frac{4\sqrt{\log_{2+}\left(\frac{d_N}{d_0}\right)}}{\sqrt{N}}.$$

$\square$

### C.3 GD Analysis

**Lemma 2.** *Assume that $d_0 \leq D$. Then, the estimate $d_k$ in Algorithm 1 satisfies $d_k \leq D$ for all $k$.*

*Proof.* By optimality of $f_*$, we have $f(x_k) - f_* \geq 0$, so

$$0 \leq \sum_{k=0}^{n} \eta_k(f(x_k) - f_*) \leq \sum_{k=0}^{n} \eta_k\langle g_k, x_k - x_*\rangle = \sum_{k=0}^{n} \eta_k\langle g_k, x_0 - x_*\rangle + \sum_{k=0}^{n} \eta_k\langle g_k, x_k - x_0\rangle.$$

Collecting the gradients in the first sum together and using Cauchy-Schwarz inequality, we obtain

$$0 \leq \sum_{k=0}^{n} \eta_k(f(x_k) - f_*) \leq \langle x_0 - x_{n+1}, x_0 - x_*\rangle + \sum_{k=0}^{n} \eta_k\langle g_k, x_k - x_0\rangle$$

$$\leq \|x_0 - x_{n+1}\|\|x_0 - x_*\| + \sum_{k=0}^{n} \eta_k\langle g_k, x_k - x_0\rangle. \tag{4}$$

Using the definition of $\hat{d}_{n+1}$, this is equivalent to $0 \leq (D - \hat{d}_{n+1})\|x_0 - x_{n+1}\|$, which implies $\hat{d}_{n+1} \leq D$. Therefore, since $d_0 \leq D$, we can show by induction $d_{n+1} \leq D$ as well. $\square$

**Lemma 3.** *The following inequality holds for the iterates of Algorithm 1:*

$$\|x_{n+1} - x_0\| \leq 2d_{n+1} + \frac{1}{2d_{n+1}} \sum_{k=0}^{n} \eta_k^2 \|g_k\|^2.$$

*Proof.* Let us rewrite $\hat{d}_{n+1}$ in a slightly different manner:

$$\hat{d}_{n+1}\|x_{n+1} - x_0\| \stackrel{\text{def}}{=} \sum_{k=0}^{n} \langle x_k - x_{k+1}, x_0 - x_k \rangle$$

$$= \sum_{k=0}^{n} \frac{1}{2} \left( \|x_{k+1} - x_0\|^2 - \|x_k - x_{k+1}\|^2 - \|x_k - x_0\|^2 \right)$$

$$= \frac{1}{2} \|x_{n+1} - x_0\|^2 - \frac{1}{2} \sum_{k=0}^{n} \|x_k - x_{k+1}\|^2.$$

Combining this with the property $\hat{d}_{n+1} \leq d_{n+1}$, we derive

$$\frac{1}{2} \|x_{n+1} - x_0\|^2 - \frac{1}{2} \sum_{k=0}^{n} \|x_k - x_{k+1}\|^2 = \hat{d}_{n+1} \|x_{n+1} - x_0\| \leq d_{n+1} \|x_{n+1} - x_0\|.$$

Applying inequality $2\alpha\beta \leq \alpha^2 + \beta^2$ with $\alpha^2 = 2d_{n+1}^2$ and $\beta^2 = \frac{1}{2}\|x_{n+1} - x_0\|^2$ and plugging-in the bound above, we establish

$$2d_{n+1}\|x_{n+1} - x_0\| = 2\alpha\beta \leq \alpha^2 + \beta^2 = 2d_{n+1}^2 + \frac{1}{2}\|x_{n+1} - x_0\|^2$$

$$\leq 2d_{n+1}^2 + d_{n+1}\|x_{n+1} - x_0\| + \frac{1}{2} \sum_{k=0}^{n} \|x_k - x_{k+1}\|^2.$$

Rearranging the terms, we obtain

$$d_{n+1}\|x_{n+1} - x_0\| \leq 2d_{n+1}^2 + \frac{1}{2} \sum_{k=0}^{n} \|x_k - x_{k+1}\|^2 = 2d_{n+1}^2 + \frac{1}{2} \sum_{k=0}^{n} \eta_k^2 \|g_k\|^2.$$

It remains to divide this inequality by $d_{n+1}$ to get the desired claim. $\qquad\square$

**Lemma 4.** *Assuming the weights $\lambda_0, \ldots, \lambda_n$ are positive, it holds for the iterates of Algorithm 1:*

$$\sum_{k=0}^{n} \frac{d_k^4 \lambda_k^2 \|g_k\|^2}{d_k^2 G^2 + \sum_{i=0}^{k} d_i^2 \lambda_i^2 \|g_i\|^2} \leq d_n^2 \log\left(1 + \sum_{k=0}^{n} \lambda_k^2\right). \tag{5}$$

*Proof.* The lemma follows straightforwardly from Proposition 2 by substituting $a_k = \frac{d_k^2}{d_n^2} \lambda_k^2 \|g_k\|^2$ for $k$ from 0 to $n$:

$$\sum_{k=0}^{n} \frac{d_k^4 \lambda_k^2 \|g_k\|^2}{d_k^2 G^2 + \sum_{i=0}^{k} d_i^2 \lambda_i^2 \|g_i\|^2} = d_n^2 \sum_{k=0}^{n} \frac{\frac{d_k^2}{d_n^2} \lambda_k^2 \|g_k\|^2}{G^2 + \sum_{i=0}^{k} \frac{d_i^2}{d_k^2} \lambda_i^2 \|g_i\|^2}$$

$$\stackrel{d_k \leq d_n}{\leq} d_n^2 \sum_{k=0}^{n} \frac{\frac{d_k^2}{d_n^2} \lambda_k^2 \|g_k\|^2}{G^2 + \sum_{i=0}^{k} \frac{d_i^2}{d_n^2} \lambda_i^2 \|g_i\|^2}$$

$$\stackrel{(3)}{\leq} d_n^2 \left( \log\left( G^2 + \sum_{k=0}^{n} \frac{d_k^2}{d_n^2} \lambda_k^2 \|g_k\|^2 \right) - \log(G^2) \right)$$

$$\leq d_n^2 \log\left(1 + \sum_{k=0}^{n} \lambda_k^2\right),$$

where in the last step we used $\frac{d_k^2}{d_n^2} \lambda_k^2 \|g_k\|^2 \leq \lambda_k^2 G^2$. $\qquad\square$

Let us restate Theorem 1:

**Theorem 7** (Same as Theorem 1). *Given any weights $1 \leq \lambda_0 \leq \cdots \lambda_n$, the functional gap of the average iterate of Algorithm 1 converges as*

$$f(\hat{x}_n) - f_* \leq \sqrt{2\lambda_n} DG \frac{2d_{n+1} + d_{n+1} \log(1 + \sum_{k=0}^n \lambda_k^2)}{\sqrt{\sum_{k=0}^n \lambda_k d_k^2}}.$$

*Proof.* The first steps in the proof follow the same lines as the theory in Defazio & Mishchenko (2023), but we still provide them for completeness.

Firstly, let us continue developing the bound proved in the proof of Lemma 2:

$$\sum_{k=0}^n \eta_k (f(x_k) - f_*) \leq \|x_0 - x_{n+1}\| D + \sum_{k=0}^n \eta_k \langle g_k, x_k - x_0 \rangle$$

$$= \|x_0 - x_{n+1}\| D + \sum_{k=0}^n \langle x_k - x_{k+1}, x_k - x_0 \rangle$$

$$= \|x_0 - x_{n+1}\| D + \frac{1}{2} \sum_{k=0}^n \left[ \|x_k - x_{k+1}\|^2 + \|x_k - x_0\|^2 - \|x_{k+1} - x_0\|^2 \right]$$

$$\leq \|x_0 - x_{n+1}\| D + \frac{1}{2} \sum_{k=0}^n \|x_k - x_{k+1}\|^2.$$

We upper bound the first term with the help of Lemma 3:

$$\sum_{k=0}^n \eta_k (f(x_k) - f_*) \leq 2Dd_{n+1} + \frac{D}{2d_{n+1}} \sum_{k=0}^n \eta_k^2 \|g_k\|^2 + \frac{1}{2} \sum_{k=0}^n \eta_k^2 \|g_k\|^2.$$

Since by Lemma 2, $1 \leq \frac{D}{d_{n+1}}$, we can simplify it to

$$\sum_{k=0}^n \eta_k (f(x_k) - f_*) \leq 2Dd_{n+1} + \frac{D}{d_{n+1}} \sum_{k=0}^n \eta_k^2 \|g_k\|^2$$

$$= 2Dd_{n+1} + \frac{D}{d_{n+1}} \sum_{k=0}^n \frac{d_k^4 \lambda_k^2}{d_k^2 G^2 + \sum_{i=0}^k d_i^2 \lambda_i^2 \|g_i\|^2} \|g_k\|^2$$

$$\overset{(5)}{\leq} 2Dd_{n+1} + \frac{D}{d_{n+1}} d_n^2 \log\left(1 + \sum_{k=0}^n \lambda_k^2\right).$$

Using the convexity of $f$, we can apply Jensen's inequality on the iterate $\hat{x}_n$ to get

$$f(\hat{x}_n) - f_* \leq \frac{1}{\sum_{k=0}^n \eta_k} \sum_{k=0}^n \eta_k (f(x_k) - f_*) \leq \frac{2Dd_{n+1} + \frac{D}{d_{n+1}} d_n^2 \log(1 + \sum_{k=0}^n \lambda_k^2)}{\sum_{k=0}^n \eta_k}$$

$$\leq D \frac{2d_{n+1} + d_{n+1} \log(1 + \sum_{k=0}^n \lambda_k^2)}{\sum_{k=0}^n \eta_k}. \tag{6}$$

Notice that

$$\eta_k = \frac{d_k^2 \lambda_k}{\sqrt{d_k^2 G^2 + \sum_{i=0}^k d_i^2 \lambda_i^2 \|g_i\|^2}} \geq \frac{d_k^2 \lambda_k}{G\sqrt{d_k^2 + \sum_{i=0}^k d_i^2 \lambda_i^2}} \geq \frac{d_k^2 \lambda_k}{G\sqrt{2\lambda_n}\sqrt{\sum_{i=0}^k d_i^2 \lambda_i}}.$$

Sum over $k$ from 0 to $n$ and using $\lambda_i \leq \lambda_n$ gives

$$\sum_{k=0}^n \eta_k \geq \frac{1}{\sqrt{2\lambda_n} G} \sum_{k=0}^n \frac{d_k^2 \lambda_k}{\sqrt{\sum_{i=0}^k d_i^2 \lambda_i}} \overset{(2)}{\geq} \frac{1}{\sqrt{2\lambda_n} G} \sqrt{\sum_{k=0}^n d_k^2 \lambda_k}.$$

Hence,

$$f(\hat{x}_n) - f_* \overset{(6)}{\leq} \sqrt{2\lambda_n} DG \frac{d_{n+1}}{\sqrt{\sum_{k=0}^n d_k^2 \lambda_k}} \left(2 + \log\left(1 + \sum_{k=0}^n \lambda_k^2\right)\right).$$

$\square$

**Corollary 1.** *Consider Algorithm 1 with $n \geq 2\log_2\left(\frac{2D}{d_0}\right)$ and define $t = \arg\min_{k \leq n} \frac{d_k}{\sqrt{\sum_{i=0}^{k} d_i^2}}$. If we choose weights $\lambda_k = 1$, then it holds*

$$f(\hat{x}_t) - f_* \leq 4\sqrt{2}DG\frac{2 + \log(n+2)}{\sqrt{n}}\sqrt{\log_2\left(\frac{2D}{d_0}\right)}.$$

*Proof.* Substituting $\lambda_k$ in the bound of Theorem 1, we get for any $n$

$$f(\hat{x}_n) - f_* \overset{(6)}{\leq} \sqrt{2}DG\frac{d_{n+1}}{\sqrt{\sum_{k=0}^{n} d_k^2}} \log(n+2).$$

Using the definition of $t$, the result of Lemma 1 and the property $d_n \leq D$, we obtain

$$f(\hat{x}_t) - f_* \leq \sqrt{2}DG\min_{k \leq n} \frac{d_{k+1}}{\sqrt{\sum_{i=0}^{k} d_i^2}}(2 + \log(n+2))$$

$$\leq 4\sqrt{2}DG\frac{2 + \log(n+2)}{\sqrt{n}}\sqrt{\log_2\left(\frac{2D}{d_0}\right)}.$$

$\square$

**Corollary 2.** *Choose any $p \geq 0$ ans set the weights to be $\lambda_k = (k+1)^p$. Then,*

$$f(\hat{x}_n) - f_* = \mathcal{O}\left(\frac{DG\sqrt{p+1}\log(n+1)}{\sqrt{n+1}}\right).$$

*Proof.* Since the sequence $d_0, d_1, \ldots$ is non-decreasing and upper bounded by $D$, there exists an index $\hat{n}$ such that $d_k \leq 2d_{\hat{n}}$ for any $k \geq \hat{n}$. Moreover, we have for $n \geq 2(\hat{n}+1)$

$$\sum_{k=\hat{n}}^{n} \lambda_k \geq \frac{1}{p+1}\left((n+1)^{p+1} - (\hat{n}+1)^{p+1}\right) \geq \frac{1}{2(p+1)}(n+1)^{p+1}$$

and

$$\sum_{k=0}^{n} \lambda_k^2 = \sum_{k=1}^{n+1} k^{2p} \leq \int_2^{n+2} x^{2p}dx \leq \frac{1}{2p+1}(n+2)^{2p+1} - 1 \leq (n+2)^{2p+1} - 1.$$

Let us plug this into the bound of Theorem 1 for $n \geq 2(\hat{n}+1)$:

$$f(\hat{x}_n) - f_* \leq \sqrt{2\lambda_n}DG\frac{d_{n+1}}{\sqrt{\sum_{k=0}^{n} d_k^2\lambda_k}}\left(2 + \log\left(1 + \sum_{k=0}^{n} \lambda_k^2\right)\right)$$

$$\leq \frac{2d_{\hat{n}}\sqrt{2(n+1)^p}DG}{\sqrt{d_{\hat{n}}^2\sum_{k=\hat{n}}^{n} \lambda_k}}(2 + (2p+1)\log(n+2))$$

$$\leq \frac{4\sqrt{p+1}DG}{\sqrt{n+1}}(2 + (2p+1)\log(n+2)) = \mathcal{O}\left(\frac{DG\sqrt{p+1}\log(n+1)}{\sqrt{n+1}}\right),$$

which matches our claim. $\square$

Notice that the bound in Corollary 2 does not depend on $D/d_0$. This is only possible asymptotically for a large enough $k$ and a similar bound without weights was presented by Defazio & Mishchenko (2023).

## C.4 DA ANALYSIS

**Lemma 5.** *Considering Algorithm 2, we have*

$$\|s_{n+1}\| \leq \frac{2d_{n+1}}{\gamma_{n+1}} + \frac{\sum_{k=0}^{n} \gamma_k \lambda_k^2 \|g_k\|^2}{2d_{n+1}}.$$

*Proof.* When studying Dual Averaging, we need to introduce an extra sequence that lower bounds $\bar{d}_n$:

$$\bar{d}_{n+1} \stackrel{\text{def}}{=} \frac{\gamma_{n+1} \|s_{n+1}\|^2 - \sum_{k=0}^{n} \gamma_k \lambda_k^2 \|g_k\|^2}{2\|s_{n+1}\|}.$$

Let us show that $\hat{d}_{n+1} \geq \bar{d}_{n+1}$ by comparing their numerators:

$$\hat{d}_{n+1}\|s_{n+1}\| = \sum_{k=0}^{n} \lambda_k \langle g_k, x_0 - x_k \rangle = \sum_{k=0}^{n} \lambda_k \gamma_k \langle g_k, s_k \rangle = \sum_{k=0}^{n} \gamma_k \langle s_{k+1} - s_k, s_k \rangle$$

$$= \sum_{k=0}^{n} \frac{\gamma_k}{2} \left[ \|s_{k+1}\|^2 - \|s_{k+1} - s_k\|^2 - \|s_k\|^2 \right]$$

$$= \frac{\gamma_n}{2} \|s_{n+1}\|^2 + \frac{1}{2} \sum_{k=0}^{n} (\gamma_k - \gamma_{k+1}) \|s_{k+1}\|^2 - \frac{1}{2} \sum_{k=0}^{n} \gamma_k \lambda_k^2 \|g_k\|^2$$

$$\stackrel{\gamma_k \geq \gamma_{k+1}}{\geq} \frac{\gamma_{n+1}}{2} \|s_{n+1}\|^2 - \frac{1}{2} \sum_{k=0}^{n} \gamma_k \lambda_k^2 \|g_k\|^2$$

$$= \bar{d}_{n+1}\|s_{n+1}\|.$$

Using the definition of $\bar{d}_{n+1}$, and the property $\bar{d}_{n+1} \leq \hat{d}_{n+1} \leq d_{n+1}$, we derive

$$\frac{\gamma_{n+1}}{2} \|s_{n+1}\|^2 - \frac{1}{2} \sum_{k=0}^{n} \gamma_k \lambda_k^2 \|g_k\|^2 = \bar{d}_{n+1} \|s_{n+1}\| \leq d_{n+1} \|s_{n+1}\|.$$

Using inequality $2\alpha\beta \leq \alpha^2 + \beta^2$ with $\alpha^2 = \frac{2d_{n+1}^2}{\gamma_{n+1}}$ and $\beta^2 = \frac{\gamma_{n+1}}{2}\|s_{n+1}\|^2$ and then the bound above, we establish

$$2d_{n+1}\|s_{n+1}\| = 2\alpha\beta \leq \alpha^2 + \beta^2 = \frac{2d_{n+1}^2}{\gamma_{n+1}} + \frac{\gamma_{n+1}}{2}\|s_{n+1}\|^2$$

$$\leq \frac{2d_{n+1}^2}{\gamma_{n+1}} + d_{n+1}\|s_{n+1}\| + \frac{1}{2} \sum_{k=0}^{n} \gamma_k \lambda_k^2 \|g_k\|^2.$$

Rearranging the terms, we obtain

$$d_{n+1}\|s_{n+1}\| \leq \frac{2d_{n+1}^2}{\gamma_{n+1}} + \frac{1}{2} \sum_{k=0}^{n} \gamma_k \lambda_k^2 \|g_k\|^2.$$

It remains to divide both sides by $d_{n+1}$. $\qquad\square$

**Lemma 6.** *The Dual Averaging algorithm (Algorithm 2) satisfies*

$$\sum_{k=0}^{n} \lambda_k (f(x_k) - f_*) \leq (D - \hat{d}_{n+1}) \|s_{n+1}\|. \tag{7}$$

*Proof.* Summing inequality $f(x_k) - f_* \leq \langle g_k, x_k - x_* \rangle$ with weights $\lambda_k$, we get

$$\sum_{k=0}^{n} \lambda_k (f(x_k) - f_*) \leq \sum_{k=0}^{n} \lambda_k \langle g_k, x_k - x_* \rangle = \sum_{k=0}^{n} \lambda_k \langle g_k, x_0 - x_* \rangle + \sum_{k=0}^{n} \lambda_k \langle g_k, x_k - x_0 \rangle.$$

Using Cauchy-Schwarz on the first product in the right-hand side and then telescoping the second sum, we obtain

$$\sum_{k=0}^{n} \lambda_k (f(x_k) - f_*) \le \|s_{n+1}\| \|x_0 - x_*\| + \sum_{k=0}^{n} \lambda_k \langle g_k, x_k - x_0 \rangle$$

$$= \|s_{n+1}\| D - \hat{d}_{n+1} \|s_{n+1}\|.$$

$\square$

Next, we restate and prove Theorem 2:

**Theorem 8** (Same as Theorem 2). *For Algorithm 2, it holds that:*

$$f(\overline{x}_t) - f_* \le \frac{4GD}{\sqrt{n}} \sqrt{\log_2 \left( \frac{2D}{d_0} \right)},$$

*where* $t = \arg\min_{k \le n} \frac{d_{k+1}}{\sqrt{\sum_{i=0}^{k} d_i^2}}$.

*Proof.* Let us sum inequality $\lambda_k (f(x_k) - f_*) \ge 0$ and then apply Lemma 6:

$$0 \le \sum_{k=0}^{n} \lambda_k (f(x_k) - f_*) \overset{(7)}{\le} (D - \hat{d}_{n+1}) \|s_{n+1}\|.$$

Clearly, this implies that $\hat{d}_{n+1} \le D$, and by induction it follows that $d_{n+1} \le D$ as well. Now let us upper bound the functional values:

$$\sum_{k=0}^{n} \lambda_k (f(x_k) - f_*) \overset{(7)}{\le} D \|s_{n+1}\| - \sum_{k=0}^{n} \gamma_k \lambda_k \langle g_k, s_k \rangle$$

$$= D \|s_{n+1}\| - \sum_{k=0}^{n} \gamma_k \langle s_{k+1} - s_k, s_k \rangle$$

$$= D \|s_{n+1}\| + \frac{1}{2} \sum_{k=0}^{n} \gamma_k \left( \|s_{k+1} - s_k\|^2 + \|s_k\|^2 - \|s_{k+1}\|^2 \right)$$

$$= D \|s_{n+1}\| + \frac{1}{2} \sum_{k=0}^{n} \gamma_k \|s_{k+1} - s_k\|^2 + \frac{1}{2} \sum_{k=0}^{n} (\gamma_k - \gamma_{k-1}) \|s_k\|^2 - \frac{\gamma_n}{2} \|s_{n+1}\|^2.$$

We can drop the last two terms since $\gamma_k \le \gamma_{k-1}$:

$$\sum_{k=0}^{n} \lambda_k (f(x_k) - f_*) \le D \|s_{n+1}\| + \frac{1}{2} \sum_{k=0}^{n} \gamma_k \|s_{k+1} - s_k\|^2$$

$$= D \|s_{n+1}\| + \frac{1}{2} \sum_{k=0}^{n} \gamma_k \lambda_k^2 \|g_k\|^2.$$

The first term in the right-hand side is readily bounded by Lemma 5:

$$\sum_{k=0}^{n} \lambda_k (f(x_k) - f_*) \le D \|s_{n+1}\| + \frac{1}{2} \sum_{k=0}^{n} \gamma_k \lambda_k^2 \|g_k\|^2$$

$$\le \frac{2D d_{n+1}}{\gamma_{n+1}} + \frac{D}{2 d_{n+1}} \sum_{k=0}^{n} \gamma_k \lambda_k^2 \|g_k\|^2 + \frac{1}{2} \sum_{k=0}^{n} \gamma_k \lambda_k^2 \|g_k\|^2$$

$$\overset{d_{n+1} \le D}{\le} \frac{2D d_{n+1}}{\gamma_{n+1}} + \frac{D}{d_{n+1}} \sum_{k=0}^{n} \gamma_k \lambda_k^2 \|g_k\|^2$$

$$\overset{\lambda_k \le \lambda_n}{\le} \frac{2D d_{n+1}}{\gamma_{n+1}} + \frac{D}{d_{n+1}} \lambda_n \sum_{k=0}^{n} \gamma_k \lambda_k \|g_k\|^2.$$

---

**Algorithm 5** Prodigy (Coordinate-wise Dual Averaging version)

1: **Input:** $d_0 > 0$, $x_0$, $G_\infty \geq 0$; $s_0 = 0 \in \mathbb{R}^p$, $a_0 = 0 \in \mathbb{R}^p$
2: **for** $k = 0$ **to** $n$ **do**
3:     $g_k \in \partial f(x_k)$
4:     $\lambda_k = d_k^2$
5:     $s_{k+1} = s_k + \lambda_k g_k$
6:     $\hat{d}_{k+1} = \dfrac{\sum_{i=0}^k \lambda_i \langle g_i, x_0 - x_i \rangle}{\|s_{k+1}\|_1}$
7:     $d_{k+1} = \max(d_k, \hat{d}_{k+1})$
8:     $a_{k+1}^2 = \lambda_{k+1} G_\infty^2 + \sum_{i=0}^k \lambda_i g_i^2$          $\triangleright$ Coordinate-wise square
9:     $\mathbf{A}_{k+1} = \mathrm{diag}(a_{k+1})$
10:     $x_{k+1} = x_k - \mathbf{A}_{k+1}^{-1} s_{k+1}$
11: **end for**
12: Return $\bar{x}_n = \frac{1}{n+1} \sum_{k=0}^n \lambda_k x_k$

---

Then, apply Proposition 1:

$$\sum_{k=0}^n \lambda_k (f(x_k) - f_*) \leq \frac{2D}{\gamma_{n+1}} + \frac{D}{d_{n+1}} \lambda_n \sum_{k=0}^n \gamma_k \lambda_k \|g_k\|^2$$

$$= \frac{2D}{\gamma_{n+1}} + \frac{D}{d_{n+1}} \lambda_n \sum_{k=0}^n \frac{1}{\sqrt{\lambda_k G^2 + \sum_{i=0}^{k-1} \lambda_i \|g_i\|^2}} \lambda_k \|g_k\|^2$$

$$\leq \frac{2D}{\gamma_{n+1}} + \frac{D}{d_{n+1}} \lambda_n \sum_{k=0}^n \frac{1}{\sqrt{\lambda_k \|g_k\|^2 + \sum_{i=0}^{k-1} \lambda_i \|g_i\|^2}} \lambda_k \|g_k\|^2$$

$$\overset{(2)}{\leq} \frac{2D}{\gamma_{n+1}} + \frac{2D}{d_{n+1}} \lambda_n \sqrt{\sum_{k=0}^n \lambda_k \|g_k\|^2}.$$

Let us now plug-in $\lambda_k = d_k^2$ and bound each gradient norm using $\|g_k\| \leq G$:

$$\sum_{k=0}^n \lambda_k (f(x_k) - f_*) \leq 4Dd_{n+1} \sqrt{\sum_{k=0}^n d_k^2 \|g_k\|^2} \leq 4GDd_{n+1} \sqrt{\sum_{k=0}^n d_k^2}.$$

Thus, we get the following convergence rate:

$$f(\bar{x}_t) - f_* \leq \frac{4GDd_{t+1} \sqrt{\sum_{k=0}^t d_k^2}}{\sum_{k=0}^t d_k^2} = \frac{4GDd_{t+1}}{\sqrt{\sum_{k=0}^t d_k^2}} = \min_{t' < n} \frac{4GDd_{t'+1}}{\sqrt{\sum_{k=0}^{t'} d_k^2}}$$

$$\leq \frac{4GD}{\sqrt{n}} \sqrt{\log_{2+}\left(\frac{D}{d_0}\right)}.$$

$\square$

## C.5   COORDINATE-WISE PRODIGY

Here we study Algorithm 5. The theory in this section follows closely the analysis in Section C.4. There are only a few minor differences such as the use of weighted norms, which we define as $\langle x, y \rangle_{\mathbf{A}^{-1}} = x^\top \mathbf{A}^{-1} y$ for any matrix $\mathbf{A} \succ 0$. In addition, we use $\ell_\infty$ norm for the distance term and for the gradients, see the assumption below.

**Assumption 2.** *The gradients are upper bounded coordinate-wise:* $\|g_k\|_\infty \leq G_\infty$.

We begin with the analogue of Lemma 5:

**Lemma 7.** *It holds for the iterates of Algorithm 5:*

$$\|s_{n+1}\|_1 \leq 2d_{n+1}\|a_{n+1}\|_1 + \frac{1}{2d_{n+1}}\sum_{k=0}^{n}\lambda_k^2\|g_k\|_{\mathbf{A}_k^{-1}}^2.$$

*Proof.* As in the proof of Lemma 5, let us introduce an extra sequence $\overline{d}_n$:

$$\overline{d}_{n+1} \stackrel{\text{def}}{=} \frac{\|s_{n+1}\|_{\mathbf{A}_{n+1}^{-1}}^2 - \sum_{k=0}^{n}\lambda_k^2\|g_k\|_{\mathbf{A}_k^{-1}}^2}{2\|s_{n+1}\|_1}.$$

The next step is to show that $\hat{d}_{n+1} \geq \overline{d}_{n+1}$ by comparing the numerators:

$$\hat{d}_{n+1}\|s_{n+1}\|_1 = \sum_{k=0}^{n}\lambda_k\langle g_k, x_0 - x_k\rangle = \sum_{k=0}^{n}\lambda_k\langle g_k, s_k\rangle_{\mathbf{A}_k^{-1}} = \sum_{k=0}^{n}\langle s_{k+1} - s_k, s_k\rangle_{\mathbf{A}_k^{-1}}$$

$$= \sum_{k=0}^{n}\frac{1}{2}\left[\|s_{k+1}\|_{\mathbf{A}_k^{-1}}^2 - \|s_{k+1} - s_k\|_{\mathbf{A}_k^{-1}}^2 - \|s_k\|_{\mathbf{A}_k^{-1}}^2\right]$$

$$= \frac{1}{2}\|s_{n+1}\|_{\mathbf{A}_n^{-1}}^2 + \frac{1}{2}\sum_{k=0}^{n}\|s_{k+1}\|_{\mathbf{A}_k^{-1}-\mathbf{A}_{k+1}^{-1}}^2 - \frac{1}{2}\sum_{k=0}^{n}\lambda_k^2\|g_k\|_{\mathbf{A}_k^{-1}}^2$$

$$\stackrel{\mathbf{A}_k^{-1}\succcurlyeq\mathbf{A}_{k+1}^{-1}}{\geq} \frac{1}{2}\|s_{n+1}\|_{\mathbf{A}_{n+1}^{-1}}^2 - \frac{1}{2}\sum_{k=0}^{n}\lambda_k^2\|g_k\|_{\mathbf{A}_k^{-1}}^2$$

$$= \overline{d}_{n+1}\|s_{n+1}\|_1.$$

Using the definition of $\overline{d}_{n+1}$, and the property $\overline{d}_{n+1} \leq \hat{d}_{n+1} \leq d_{n+1}$, we derive

$$\frac{1}{2}\|s_{n+1}\|_{\mathbf{A}_{n+1}^{-1}}^2 - \frac{1}{2}\sum_{k=0}^{n}\lambda_k^2\|g_k\|_{\mathbf{A}_k^{-1}}^2 = \overline{d}_{n+1}\|s_{n+1}\|_1 \leq d_{n+1}\|s_{n+1}\|_1.$$

Using inequality $2\alpha\beta \leq \alpha^2 + \beta^2$ with $\alpha^2 = 2d_{n+1}^2 a_{(n+1)i}$ and $\beta^2 = \frac{1}{2a_{(n+1)i}}s_{(n+1)i}^2$ for $i = 1, \ldots, p$ and then the bound above, we establish

$$2d_{n+1}\|s_{n+1}\|_1 = \sum_{i=1}^{p}d_{n+1}|s_{(n+1)i}| \leq \sum_{i=1}^{p}\left(2d_{n+1}^2 a_{(n+1)i} + \frac{1}{2a_{(n+1)i}}s_{(n+1)i}^2\right)$$

$$= 2d_{n+1}^2\|a_{n+1}\|_1 + \frac{1}{2}\|s_{n+1}\|_{\mathbf{A}_{n+1}^{-1}}$$

$$\leq 2d_{n+1}^2\|a_{n+1}\|_1 + d_{n+1}\|s_{n+1}\|_1 + \frac{1}{2}\sum_{k=0}^{n}\lambda_k^2\|g_k\|_{\mathbf{A}_k^{-1}}^2.$$

Rearranging the terms, we get

$$d_{n+1}\|s_{n+1}\|_1 \leq 2d_{n+1}^2\|a_{n+1}\|_1 + \frac{1}{2}\sum_{k=0}^{n}\lambda_k^2\|g_k\|_{\mathbf{A}_k^{-1}}^2.$$

It remains to divide both sides by $d_{n+1}$. □

The next lemma is similar to Lemma 7 except that it uses $\ell_\infty$ norm for the distance to a solution and $\ell_1$ norm for the weighted gradient sum $s_n$.

**Lemma 8.** *The coordinate-wise version of Prodigy (Algorithm 5) satisfies*

$$\sum_{k=0}^{n}\lambda_k(f(x_k) - f_*) \leq (D_\infty - \hat{d}_{n+1})\|s_{n+1}\|_1, \tag{8}$$

*where $D_\infty = \|x_0 - x_*\|_\infty$.*

*Proof.* Summing inequality $f(x_k) - f_* \leq \langle g_k, x_k - x_* \rangle$ with weights $\lambda_k$, we get

$$\sum_{k=0}^{n} \lambda_k (f(x_k) - f_*) \leq \sum_{k=0}^{n} \lambda_k \langle g_k, x_k - x_* \rangle = \sum_{k=0}^{n} \lambda_k \langle g_k, x_0 - x_* \rangle + \sum_{k=0}^{n} \lambda_k \langle g_k, x_k - x_0 \rangle.$$

Using Hölder's inequality on the first product in the right-hand side and then telescoping the second sum, we obtain

$$\sum_{k=0}^{n} \lambda_k (f(x_k) - f_*) \leq \|s_{n+1}\|_1 \|x_0 - x_*\|_\infty + \sum_{k=0}^{n} \lambda_k \langle g_k, x_k - x_0 \rangle$$

$$= \|s_{n+1}\|_1 D_\infty - \hat{d}_{n+1} \|s_{n+1}\|.$$

The use of $\ell_1$ norm for the term $s_{n+1}$ above is motivated by the fact that it naturally arises in other parts of the theory. $\qquad\square$

**Theorem 9.** *Algorithm 5 converges with the rate*

$$f(\overline{x}_t) - f_* \leq \frac{4pG_\infty D_\infty}{\sqrt{n}} \sqrt{\log_{2+}\left(\frac{D_\infty}{d_0}\right)},$$

*where* $t = \arg\min_{k \leq n} \frac{d_{k+1}}{\sqrt{\sum_{i=0}^{k} d_i^2}}$.

*Proof.* From Lemma 8, we get

$$0 \leq \sum_{k=0}^{n} \lambda_k (f(x_k) - f_*) \overset{(8)}{\leq} (D_\infty - \hat{d}_{n+1}) \|s_{n+1}\|_1,$$

so we can prove by induction that $d_{n+1} \leq D_\infty$. Using the same bounds as before, we get for the average iterate

$$\sum_{k=0}^{n} \lambda_k (f(x_k) - f_*) \leq D_\infty \|s_{n+1}\|_1 - \sum_{k=0}^{n} \lambda_k \langle g_k, x_0 - x_k \rangle$$

$$= D_\infty \|s_{n+1}\|_1 + \frac{1}{2} \sum_{k=0}^{n} \lambda_k^2 \|g_k\|_{\mathbf{A}_k^{-1}}^2 + \frac{1}{2} \sum_{k=0}^{n} \|s_k\|_{\mathbf{A}_k^{-1} - \mathbf{A}_{k+1}^{-1}}^2 - \frac{1}{2} \|s_{n+1}\|_{\mathbf{A}_{n+1}^{-1}}^2$$

$$\leq D_\infty \|s_{n+1}\|_1 + \frac{1}{2} \sum_{k=0}^{n} \lambda_k^2 \|g_k\|_{\mathbf{A}_k^{-1}}^2.$$

Let us plug in the bound from Lemma 7:

$$\sum_{k=0}^{n} \lambda_k (f(x_k) - f_*) \leq 2D_\infty d_{n+1} \|a_{n+1}\|_1 + \frac{D_\infty}{2d_{n+1}} \sum_{k=0}^{n} \lambda_k^2 \|g_k\|_{\mathbf{A}_k^{-1}}^2 + \frac{1}{2} \sum_{k=0}^{n} \lambda_k^2 \|g_k\|_{\mathbf{A}_k^{-1}}^2$$

$$\overset{d_{n+1} \leq D_\infty}{\leq} 2D_\infty d_{n+1} \|a_{n+1}\|_1 + \frac{D_\infty}{d_{n+1}} \sum_{k=0}^{n} \lambda_k^2 \|g_k\|_{\mathbf{A}_k^{-1}}^2$$

$$\overset{\lambda_k \leq \lambda_n}{\leq} 2D_\infty d_{n+1} \|a_{n+1}\|_1 + \frac{D_\infty}{d_{n+1}} \lambda_n \sum_{k=0}^{n} \lambda_k \|g_k\|_{\mathbf{A}_k^{-1}}^2.$$

We now apply Proposition 1, substitute $\lambda_k = d_k^2$, and use $g_{kj}^2 \leq G_\infty^2$:

$$\sum_{k=0}^{n} d_k^2 (f(x_k) - f_*) \leq 2D_\infty d_{n+1} \|a_{n+1}\|_1 + \frac{D_\infty}{d_{n+1}} \lambda_n \sum_{j=1}^{p} \sum_{k=0}^{n} \frac{\lambda_k g_{kj}^2}{\sqrt{d_k^2 G_\infty^2 + \sum_{i=0}^{k-1} \lambda_i g_{ij}^2}}$$

$$\leq 2D_\infty d_{n+1} \|a_{n+1}\|_1 + \frac{2D_\infty}{d_{n+1}} \lambda_n \sum_{j=1}^{p} \sqrt{\sum_{k=0}^{n} \lambda_k g_{kj}^2}$$

$$\leq 4D_\infty d_{n+1} pG_\infty \sqrt{\sum_{k=0}^{n} d_k^2}.$$

Using Lemma 1, we get the rate for $t = \arg\min_{t' \leq n} \frac{d_{t'+1}}{\sqrt{\sum_{k=0}^{t'} d_k^2}}$:

$$f(\overline{x}_t) - f_* \leq \frac{4pG_\infty D_\infty}{\sqrt{n}} \sqrt{\log_{2+}\left(\frac{D_\infty}{d_0}\right)}.$$

$\square$

# D    ANALYSIS OF D-ADAPTATION WITH RESETTING

In Algorithm 3 the $r$ counter tracks the epoch. Let $n_r$ represent the number of steps performed in epoch $r$. Let $R \leq \log_{2+}(D/d_0)$ denote the total number of epochs performed before the algorithm returns, where $D = \|x_0 - x_*\|$.

**Lemma 9.** *Consider the steps within a single epoch, dropping the $r$ index, we have that the norm of $s_{n+1}$ is bounded by:*

$$\|s_{n+1}\| \leq 5G\sqrt{n+1}. \tag{9}$$

*Proof.* We start with Lemma 5 from Defazio & Mishchenko (2023), which applies within an epoch in our case since the gamma decreases within epochs:

$$-\sum_{k=0}^{n} \gamma_k \langle g_k, s_k \rangle \leq -\frac{\gamma_n}{2} \|s_{n+1}\|^2 + \sum_{k=0}^{n} \frac{\gamma_k}{2} \|g_k\|^2.$$

Note that we have used a slightly tightened version, where $\gamma_n$ rather than $\gamma_{n+1}$, appears on the right, which easily follows by not using $\gamma_{n+1} \leq \gamma_n$ on the last step of their telescoping.

Using the definition of $\hat{d}_{n+1}$ and the property $\hat{d}_{n+1} \leq 2d$, we have:

$$\frac{\gamma_n}{2} \|s_{n+1}\|^2 - \sum_{k=0}^{n} \frac{\gamma_k}{2} \|g_k\|^2 = \hat{d}_{n+1} \|s_{n+1}\| \leq 2d \|s_{n+1}\|.$$

Using inequality $2\alpha\beta \leq \alpha^2 + \beta^2$ with $\alpha^2 = \frac{4d^2}{\gamma_n}$ and $\beta^2 = \frac{\gamma_n}{4} \|s_{n+1}\|^2$ and then the bound above, we establish

$$2\alpha\beta = 2d\|s_{n+1}\| \leq \frac{4d^2}{\gamma_n} + \frac{\gamma_n}{4} \|s_{n+1}\|^2$$

$$\leq \frac{4d^2}{\gamma_n} + d\|s_{n+1}\| + \frac{1}{2} \sum_{k=0}^{n} \gamma_k \|g_k\|^2.$$

Rearranging the terms, we obtain

$$d\|s_{n+1}\| \leq \frac{4d^2}{\gamma_n} + \sum_{k=0}^{n} \frac{\gamma_k}{2} \|g_k\|^2.$$

It remains to divide this inequality by $d$ to get:

$$\|s_{n+1}\| \leq \frac{4d}{\gamma_n} + \frac{1}{d} \sum_{k=0}^{n} \frac{\gamma_k}{2} \|g_k\|^2.$$

Now plugging in $\gamma_n = d/\sqrt{G^2 + \sum_{k=0}^{n-1} \|g_k\|^2}$, and using the AdaGradNorm error bound:

$$\|s_{n+1}\| \leq 4\sqrt{G^2 + \sum_{k=0}^{n} \|g_k\|^2} + \frac{1}{2} \sum_{k=0}^{n} \frac{\|g_k\|^2}{\sqrt{G^2 + \sum_{i=0}^{k-1} \|g_i\|^2}}$$

$$\leq 4\sqrt{G^2 + \sum_{k=0}^{n-1} \|g_k\|^2} + \sqrt{\sum_{k=0}^{n} \|g_k\|^2}$$

$$\leq 5G\sqrt{n+1}.$$

$\square$

**Lemma 10.** *For epoch $r$ we have:*

$$\sum_{k=0}^{n_r} \left( f(x_{k,r}) - f_* \right) \leq 6DG\sqrt{n_r + 1}.$$

*Proof.* Starting from the bound in Defazio & Mishchenko (2023):

$$\sum_{k=0}^{n_r} \left( f(x_{k,r}) - f_* \right) \leq D \left\| s_{n+1,r} \right\| + \frac{1}{2} \sum_{k=0}^{n_r} \gamma_{k,r} \left\| g_{k,r} \right\|^2 - \frac{1}{2} \gamma_{n_r+1,r} \left\| s_{n_r+1,r} \right\|^2 .$$

Then we apply Lemma 9:

$$\sum_{k=0}^{n_r} \left( f(x_{k,r}) - f_* \right) \leq 5DG\sqrt{n_r + 1} + \frac{1}{2} \sum_{k=0}^{n_r} \gamma_{k,r} \left\| g_{k,r} \right\|^2 - \frac{1}{2} \gamma_{n_r+1,r} \left\| s_{n_r+1,r} \right\|^2 .$$

Using the AdaGradNorm bound $\frac{1}{2} \sum_{k=0}^{n_r} \gamma_{k,r} \left\| g_{k,r} \right\|^2 \leq D\sqrt{\sum_{k=0}^{n_r} \left\| g_{r,k} \right\|^2} \leq DG\sqrt{n_r + 1}$. and further dropping the $-\frac{1}{2} \gamma_{n_r+1,r} \left\| s_{n_r+1,r} \right\|^2$ term gives the result. $\square$

**Theorem 10.** *For Algorithm 3, it holds that:*

$$f(\bar{x}_n) - f_* \leq \frac{6DG\sqrt{\log_{2+}(D/d_0)}}{\sqrt{n+1}}.$$

*Proof.* Starting from Lemma 10, for epoch $r$ it holds that:

$$\sum_{k=0}^{n_r} \left( f(x_{k,r}) - f_* \right) \leq 6DG\sqrt{n_r + 1}.$$

We sum over epochs up to the final epoch $R$:

$$\sum_{k=0}^{n} \left( f(x_k) - f_* \right) \leq 6DG \sum_{r=1}^{R} \sqrt{n_r + 1}.$$

Jensen's inequality tells us that for concave $\varphi$:

$$\frac{\sum_{r=1}^{R} a_r \varphi(x_r)}{\sum_{r=1}^{R} a_r} \leq \varphi \left( \frac{\sum_{r=1}^{R} a_r x_r}{\sum_{r=1}^{R} a_r} \right).$$

Applying to our case, we use $\varphi(x) = \sqrt{x}$, $a_r = 1$, $\sum_{r=1}^{R} a_r = R$, and $x_r = n_{r+1}$:

$$\frac{\sum_{r=1}^{R} \varphi(x_r)}{R} \leq \varphi \left( \frac{\sum_{r=1}^{R} n_{r+1}}{R} \right),$$

so

$$\frac{\sum_{r=1}^{R} \sqrt{n_r + 1}}{R} \leq \sqrt{\frac{n+1}{R}},$$

which we rearrange into

$$\therefore \sum_{r=1}^{R} \sqrt{n_r + 1} \leq \sqrt{R(n+1)}.$$

Noting that $R \leq \log_{2+}(D/d_0)$:

$$\sum_{k=0}^{n} \left( f(x_k) - f_* \right) \leq 6DG\sqrt{(n+1)\log_{2+}(D/d_0)}.$$

Applying Jensen's inequality to obtain a bound on the average iterate completes the proof. $\square$

# E  LOWER COMPLEXITY THEORY

**Theorem 11.** *Consider any Algorithm for minimizing a convex $G$-Lipschitz function starting from $x_0$ at the origin, which has no knowledge of problem constants. At each iteration $k$, the algorithm may query the gradient at a single point $x_k$. Then for any sequence of $x_{1,...}, x_n$, there exists a convex Lipschitz problem $f$ and constant $D \geq \|x_0 - x_*\|$ for all minimizers $x_*$ of $f$ such that:*

$$\min_{k \leq n} f(x_k) - f_* \geq \frac{DG\sqrt{\log_2 \log_2 (D/x_1)}}{2\sqrt{n+1}}.$$

*Proof.* Firstly let $g_0 = -1$. We consider two cases. Case 1) Suppose that $x_k \leq \frac{1}{2} 2^{2^{n+1}} x_1$ for all $k$. Then define

$$x_* = 2^{2^{n+1}} x_1,$$

so that $|x_0 - x_*| = D = 2^{2^{n+1}} x_1$. Our construction uses the gradient sequence $g_k = -1$ for all k. This corresponds to the function:

$$f(x) = |x - x_*|.$$

Note that for all query points $x$, the gradient is negative, and only the left arm of the absolute value function is seen by the algorithm, so the function appears linear for all test points. Using this construction, we have:

$$\min_{k \leq n} [f(x_k) - f_*] = \min_{k \leq n} (x_* - x_k)$$
$$= 2^{2^{n+1}} x_1 - \max_{k \leq n} x_k$$
$$\geq 2^{2^{n+1}} x_1 - \frac{1}{2} 2^{2^{n+1}} x_1$$
$$\geq \frac{1}{2} D_n.$$

Now note that:

$$\sqrt{\log \log_2 (D/x_1)} = \sqrt{\log_2 \log_2 (2^{2^{n+1}})} = \sqrt{n+1}.$$

So combining these two results:

$$\min_{k \leq n} f(x_k) - f_* \geq \frac{1}{2} DG$$
$$= \frac{DG\sqrt{\log_2 \log_2 (D/x_1)}}{2\sqrt{n+1}}.$$

Case 2). Otherwise, there exists a $k$ such that $x_{k+1} \geq \frac{1}{2} 2^{2^{n+1}} x_1$. Then we will fix $x_*$ to be in the interval $I = [2^{2^{n+1}} x_1, 2^{2^{n+2}} x_1]$. Our gradient oracle will return $g(x_k) = \text{sign}(x_k - x_*)$, and the corresponding function is $f(x) = |x - x_*|$. Since the problem can only become harder with less information, we may assume that the algorithm knows the end points of the interval I, and we treat $x_k$ as the first query point in the interval. Without loss of generality we further assume $k = 2$ since any larger value just gives fewer query points and thus a worse bound.

For the resisting oracle, we can use the same resisting oracle as would be used for binary search, but applied to the logarithm of the points. For root finding on an interval $[a, b]$ the lower complexity bound for $t$ steps is know to be (Sikorski, 1982):

$$|x - x_*| \geq \frac{1}{2^{t+1}} |b - a|.$$

and so since we are taking $n - 1$ steps (since we start at $x_2$):

$$|\log_2 x_n - \log_2 x_*| \geq \frac{1}{2^n} \left( \log_2 2^{2^{n+2}} x_1 - \log_2 2^{2^{n+1}} x_1 \right)$$

$$\geq \frac{1}{2^n} \left( 2^{n+2} - 2^{n+1} \right)$$

$$= \frac{1}{2^n} \left( 2 \cdot 2^{n+1} - 2^{n+1} \right) = \frac{2^{n+1}}{2^n}$$

$$= 2.$$

Therefore either $x_n \leq \frac{1}{4} x_*$ or $x_n \geq 4 x_*$. Therefore:

$$f(x_k) - f_* \geq \min \left\{ \left| \frac{1}{4} x_* - x_* \right|, |4 x_* - x_*| \right\}$$

$$= \frac{3}{4} x_* = \frac{3}{4} DG.$$

Note that $D \leq 2^{2^{n+2}} x_1$, so:

$$\sqrt{\log \log_2(D/x_1)} \leq \sqrt{\log_2 \log_2(2^{2^{n+2}})} = \sqrt{n+2},$$

$$\text{therefore, } 1 \geq \frac{\sqrt{\log \log_2(D/x_1)}}{\sqrt{n+2}},$$

and so multiplying the two bounds gives:

$$\min_{k \leq n} f(x_k) - f_* \geq \frac{3 DG \sqrt{\log_2 \log_2(D/x_1)}}{4\sqrt{n+2}}$$

$$\geq \frac{DG \sqrt{\log_2 \log_2(D/x_1)}}{2\sqrt{n+1}}.$$

$\square$

**Theorem 12.** *Consider any exponentially bounded algorithm for minimizing a convex $G$-Lipschitz function starting from $x_0$, which has no knowledge of problem constants $G$ and $D$. There exists a fixed gradient oracle such that any sequence of $x_{1,\ldots,}, x_n$, there exists a convex Lipschitz problem $f$ with $G = 1$ and $\|x_0 - x_*\| \leq D$ for all minimizing points $x_*$, consistent with the gradient oracle such that:*

$$\min_{k \leq n} f(x_k) - f_* \geq \frac{DG \sqrt{\log_2(D/x_1)}}{2\sqrt{n+1}}.$$

*Proof.* We consider the construction of a 1D oracle for this problem. Our oracle returns $g_0 = -1$ and $f(x_k) = -x_k$ for all queries. Without loss of generality we assume that $x_k > 0$ for all $k \geq 1$, and $G = 1$.

For each step $k \geq 1$ we define:

$$x_* = 2^{n+1} x_1,$$

and thus $D = |x_0 - x_*| = 2^{k+1} x_1$. and our construction uses the following function value and gradient sequence

$$f(x) = |x - x_*| + x_*.$$

Note that for all query points $x$, the gradient is negative, and only the left arm of the absolute value function is seen by the algorithm, so the function appears linear for all test points. Using this construction, we have:

$$\min_{k \leq n} [f(x_k) - f_*] = \min_{k \leq n} (x_* - x_k)$$

$$= 2^{n+1} x_1 - \max_{k \leq n} x_k$$

$$\geq 2 \cdot 2^n x_1 - 2^n x_1$$

$$= 2^n x_1$$

$$= \frac{1}{2} D_n.$$

Now note that:

$$\sqrt{\log_2(D_n/x_1)} = \sqrt{\log_2(2^{n+1})}$$
$$= \sqrt{n+1}.$$

So:

$$1 \geq \frac{\sqrt{\log_2(D_n/x_1)}}{\sqrt{n+1}}.$$

Combining these two results:

$$\min_{k \leq n} f(x_k) - f_* \geq \frac{1}{2}D = \frac{1}{2}DG$$
$$= \frac{\frac{1}{2}DG\sqrt{\log_2(D/x_1)}}{\sqrt{n+1}}.$$

$\square$

**Theorem 13.** *D-Adaptation, DoG, Prodigy and D-Adaptation with resetting are exponentially bounded.*

*Proof.* Consider the $D$ lower bound from D-Adaptation:

$$\hat{d}_{n+1} = \frac{\sum_{k=0}^{n} \lambda_k \gamma_k \langle g_k, s_k \rangle}{\|s_{n+1}\|},$$

with:

$$s_{n+1} = \sum_{k=0}^{n} d_k g_k.$$

Recall that

$$\sum_{k=0}^{n} \lambda_k \gamma_k \langle g_k, s_k \rangle \leq \gamma_{n+1} \|s_{n+1}\|^2.$$

Note also that $\gamma_{n+1} \leq \frac{1}{G}$. So:

$$d_{n+1} \leq \frac{\frac{1}{G}\|s_{n+1}\|^2}{\|s_{n+1}\|} \leq \frac{1}{G}\left\|\sum_{k=0}^{n} d_k g_k\right\| \leq \sum_{k=0}^{n} d_k.$$

So the sequence $d_n$ is upper bounded by the sequence:

$$a_n = \begin{cases} \sum_{k=0}^{n-1} a_k & n \geq 1 \\ d_0 & n = 0 \end{cases}.$$

This sequence has the following closed form:

$$a_{n+1} = 2^n d_0 \quad \text{for } n \geq 1.$$

We can prove this by induction. The base case is by definition $a_1 = a_0$. Then

$$a_{n+1} = \sum_{k=0}^{n} a_k = \sum_{k=0}^{n-1} a_k + a_n = a_n + a_n = 2a_n = 2^n d_0.$$

Note that for both the Dual Averaging form and the GD form we have, we have:

$$\|x_{n+1} - x_0\| \leq \left\|\frac{1}{G}\sum_{k=0}^{n} d_k g_k\right\| \leq \sum_{k=0}^{n} d_k \leq d_{n+1} \leq 2^n d_0.$$

It follows that D-Adaptation is exponentially bounded. The same argument applies to the resetting variant since the resetting operation does not increase the rate of accumulation of $d$.

For Prodigy, note that:

$$\gamma_{n+1} \leq \frac{1}{\sqrt{d_{n+1}^2 G^2}} = \frac{1}{d_{n+1} G}.$$

Therefore

$$d_{n+1} \leq \frac{\frac{1}{d_{n+1}G} \|s_{n+1}\|^2}{\|s_{n+1}\|} \leq \frac{1}{d_{n+1}G} \left\| \sum_{k=0}^{n} d_k^2 g_k \right\| \leq \frac{1}{d_{n+1}} \sum_{k=0}^{n} d_k^2 \leq \frac{1}{d_{n+1}} \sum_{k=0}^{n} d_k d_{n+1}$$

$$\leq \sum_{k=0}^{n} d_k.$$

The rest of the argument follows the D-Adaptation case, with:

$$\|x_{n+1} - x_0\| \leq \left\| \frac{1}{d_n G} \sum_{k=0}^{n} d_k^2 g_k \right\| \leq \sum_{k=0}^{n} d_k \leq d_{n+1} \leq 2^n d_0.$$

For DoG, recall the basic DoG step is gradient descent with step sizes:

$$\gamma_k = \frac{\bar{r}_k}{\sqrt{G^2 + \sum_{i=0}^{k} \|g_i\|^2}}.$$

Using the triangle inequality we have:

$$\begin{aligned}
\|x_{k+1} - x_0\| &= \|x_k - \gamma_k g_k - x_0\| \\
&\leq \|x_k - x_0\| + \gamma_k \|g_k\| \\
&\leq \|x_k - x_0\| + \frac{\bar{r}_k}{\sqrt{G^2}} \|g_k\| \\
&\leq \|x_k - x_0\| + \bar{r}_k \\
&\leq 2\bar{r}_k.
\end{aligned}$$

Chaining gives the result.

$\square$

**Proposition 3.** *Suppose that $d_k \leq cD$ and $\gamma_k \leq d_k/G$. then:*
$$\|x_k - x_0\| \leq (2c+1)^n \|x_1 - x_0\|.$$

*Proof.* Without loss of generality assume that $G = 1$. Firstly, note that using the absolute value function as constructed in Theorem 6, it's clear that there is always exists a function with $D_k \leq 2\|x_k - x_*\|$ at step $k$ consistent with the sequence of gradients seen so far. Therefore, it must hold that

$$d_k \leq cD_k \leq 2c\|x_k - x_0\|.$$

We prove the result by induction. For the base case, trivially:

$$\|x_1 - x_0\| \leq (2c+1)^1 \|x_1 - x_0\|.$$

For the inductive case:

$$\begin{aligned}
\|x_{k+1} - x_0\| &= \|x_k - \gamma_k g_k - x_0\| \\
&\leq \|x_k - x_0\| + \gamma_k \|g_k\| \\
&\leq \|x_k - x_0\| + \frac{cD_k}{G} \|g_k\| \\
&\leq \|x_k - x_0\| + cD_k \\
&\leq (2c+1) \|x_k - x_0\| \\
&\leq (2c+1)^{n+1} \|x_1 - x_0\|.
\end{aligned}$$

$\square$

