# OpenReview forum: "Prodigy: An Expeditiously Adaptive Parameter-Free Learner"
_ICLR.cc/2024/Conference — Submitted to ICLR 2024_

### Official Review · Reviewer_gz9E · 2023-10-29

**Soundness:** 1 poor
**Presentation:** 1 poor
**Contribution:** 1 poor
**Rating:** 1
**Confidence:** 5

**Summary:**

The paper develops an improvement to the D-Adaptation algorithm of Defazio & Mischenko (2023), improving the rate from $O\left(\frac{\\|x_0-x^\*\\|\log\left(\\|x_0-x^\*\\|/d_0\right)}{\sqrt{T}}\right)$  to $O\left(\frac{\\|x_0-x^\*\\|\sqrt{\log\left(\\|x_0-x^\*\\|/d_0\right)}}{\sqrt{T}}\right)$.

**Strengths:**

The paper is easy to read and mostly free of typos and grammatical errors.

**Weaknesses:**

## Summary
Overall, I very strongly recommend rejection. The paper makes no new theoretical contributions and is seemingly unaware of large portions of the related literature. The results presented here have already been achieved more generally and in harder problem settings, and the lower bounds are both invalid. The experiments show some evidence of improvement over D-Adaptation, but makes no attempt to compare against the obvious existing baselines from the online learning literature. The main novelty is the improvement of the results from the D-Adaptation paper, which suffered from all of the same issues mentioned above, so I don't believe improving over this work warrants publication. Additional discussion on these points is provided below.

## The Algorithm
Algorithms that attain the $\\|x_0-x^\*\\|\sqrt{\log(\\|x_0-x^\*\\|/d_0)}/\sqrt{T}$ rate (or equivalently $\\|x_0-x^\*\\|\sqrt{T\log(\\|x_0-x^\*\\|/d_0)}$ regret) under Lipschitz losses have existed for almost a decade now, and they accomplish it in *strictly* harder problems (see e.g. McMahan & Orabona 2014, Orabona & Pal 2016, Cutkosky & Orabona 2018). In particular, there are countless algorithms from the  so-called "parameter-free" online learning literature which already achieve this result in the significantly harder adversarial online learning setting, in which the $\\|x_0-x^*\\|\sqrt{T\log(\\|x_0-x^\*\\|/d_0)}$ regret is un-improvable. The methods presented here achieve this rate in the *easiest possible problem setting*: optimizing a *fixed* function, where this rate isn't even optimal. Non-trivial extensions have even been achieved in these harder problem settings, such as as scale-free learning (Mhammedi & Koolen 2020), dynamic regret (Jacobsen & Cutkosky 2022), learning with switching costs (Zhang & Cutkosky 2022b), and many more. Note that these results have been achieved using a variety of approaches: coin-betting, FTRL, mirror descent, and general potential-based approaches, while the discussion in Section 5 seems to only be aware of the coin-betting approach.

The paper also claims to improve on methods other than D-Adaptation, such as the T-DoG algorithm of Ivgi et al (2023). This is factually incorrect: the problem setting studied in this paper is strictly easier than all other comparable works in this field (besides D-Adaptation), so Prodigy actually has *no* guarantee in the problem setting studied by Ivgi et al. 2023. In this sense Prodigy is actually a *strict downgrade* of T-DoG, and likewise of any of the other existing works that can solve the problem studied here.

Section 5 also implies that standard D-adaptation (and Prodigy by extension) actually improve over the existing online learning works, as "Standard D-Adaptation obtains asymptotic rates without the log factor". This is at least misleading: their asymptotic result holds under the condition that the **user** chooses $d_0\le \\|x_0-x^\*\\|$, which would only be possible to guarantee if you have prior knowledge of $\\|x_0-x^\*\\|$. If you had this prior knowledge, you wouldn't need any special algorithm to achieve the $O(G\\|x_0-x^\*\\|/\sqrt{T})$ rate, you could accomplish this using gradient descent with step-sizes $\eta_t = \frac{\\|x_0-x^\*\\|}{\sqrt{G^2+\sum_s^{t-1}\\|g_s\\|^2}}$. One might argue that you can just set $d_0$ to be very close to 0, but the smaller you set $d_0$, the longer it takes for the asymptotic result to kick in, making it easy to wind up with a result that holds only for some $\tilde T$ such that $G\\|x_0-x^\*\\|\sqrt{\tilde T}\ge G\\|x_0-x^\*\\|\sqrt{T\log(\\|x^\*-x_0\\|/d_0)}$, making the result again redundant. So there is no meaningful improvement via the asymptotic result either.

An Adam-based variant is also proposed, but this leads to something of a contradiction: Adam itself isn't guaranteed to converge (Reddi et al. 2019), so Prodigy+Adam isn't guaranteed to converge either. How can we claim an algorithm as "parameter-free" if it makes zero performance guarantees and may diverge? The whole point is that the algorithm makes some performance *guarantee* without tuning hyperparameters. Moreover, it's unclear to me how interesting the global step-size of Adam even is --- the momentum parameters have far more impact on the behavior of Adam. The experiments claim that Prodigy+Adam performs roughly as well as hand-tuned Adam, but I would not be surprised if Adam with the default global step-size *also* performed similarly to hand-tuned Adam.

## The Lower bounds
Neither of the lower bounds are valid constructions. The lower bounds choose $D$ to be either $2^{2^T}x_1$ or $2^Tx_1$, neither of which are valid ways to construct the stated lower bound, because now the bound holds for only a specific class of comparators, rather than for all comparators simultaneously. In other words, the lower bounds should show that *for any* $D,G$, there is an $x^\*$ such that $\\|x_0-x^\*\\|= D$ and the stated lower bound holds. This is also what the theorem statement *implies* is happening, until you actually read the proof, which is misleading.

## The Experiments
The experiments demonstrate some improvement over D-Adaptation, but are not particularly convincing aside from that. Notably, the experiments include *no* baselines from the many existing works from online learning. Given that there are several existing algorithms that already achieve stronger results than presented here, this paper should at the *very least* be justifying its existence by showing some improvement over these existing works. Yet not a single one is included as a baseline.

The experiments use other tricks on top of Prodigy such as a warm-up epoch, step-size annealing, and weight-decay, so I'm again unsure how the algorithm can be claimed to be an "Expeditiously Adaptive Parameter-Free Learner" --- these all involve some form of hyperparameter selection. I also don't understand why weight decay is necessary; adding an L2 penalty to the loss will implicitly constrain the algorithm to a ball of some radius, which shouldn't even be necessary for an algorithm that already adapts to $\\|x_0-x^\*\\|$. The fact that this needed to be included suggests that the algorithm in fact does *not* attain this form of adaptivity in general, as one would hope to demonstrate in the experiments.

The Large-scale Adam experiments of Section 6.1 show evidence that the performance of Prodigy can be similar to that of hand-tuned Adam. However, as mentioned earlier, this is not particularly convincing on its own because Adam's global step-size has a relatively benign impact on performance. These experiments should at least include "un-tuned Adam" using the default step-size, which I suspect will also perform similarly to Prodigy.

## References
- Cutkosky, Ashok, and Francesco Orabona. "Black-box reductions for parameter-free online learning in banach spaces." Conference On Learning Theory. PMLR, 2018.
- Mhammedi, Zakaria, and Wouter M. Koolen. "Lipschitz and comparator-norm adaptivity in online learning." Conference on Learning Theory. PMLR, 2020.
- Jacobsen, Andrew, and Ashok Cutkosky. "Parameter-free mirror descent." Conference on Learning Theory. PMLR, 2022.
- Orabona, Francesco, and Dávid Pál. "Parameter-free stochastic optimization of variationally coherent functions." arXiv preprint arXiv:2102.00236 (2021).
- Orabona, Francesco. "A modern introduction to online learning." arXiv preprint arXiv:1912.13213 (2019).
- Orabona, Francesco, and Dávid Pál. "Coin betting and parameter-free online learning." Advances in Neural Information Processing Systems 29 (2016).
- McMahan, H. Brendan, and Francesco Orabona. "Unconstrained online linear learning in hilbert spaces: Minimax algorithms and normal approximations." Conference on Learning Theory. PMLR, 2014.
- McMahan, Brendan, and Matthew Streeter. "No-regret algorithms for unconstrained online convex optimization." Advances in neural information processing systems 25 (2012).
- Reddi, Sashank J., Satyen Kale, and Sanjiv Kumar. "On the convergence of adam and beyond." arXiv preprint arXiv:1904.09237 (2019).
- Zhang, Jiujia, and Ashok Cutkosky. "Parameter-free regret in high probability with heavy tails." Advances in Neural Information Processing Systems 35 (2022a).
- Zhang, Zhiyu, Ashok Cutkosky, and Yannis Paschalidis. "Optimal Comparator Adaptive Online Learning with Switching Cost." Advances in Neural Information Processing Systems 35 (2022b)

**Questions:**

- What new contributions does this work make that haven't already been addressed in the online learning literature?
- Why do the experiments include no baselines from the relevant online learning literature?

---

> ### Author Response · Authors · 2023-11-22
>
> The reviewer gave us the lowest possible score with the highest possible confidence, and made a few very strong claims in their review such as “The paper makes no new theoretical contributions”, “there is no meaningful improvement”, “it's unclear to me how interesting the global step-size of Adam even is”, etc. Moreover, as the reviewer pointed out themself, all of the claims in the review apply to the D-Adaptation paper, which received the outstanding paper award from ICML, but the reviewer would probably have given the same score to that paper too. Since the reviewer has a strong opinion on the prior literature that might be affecting this review more than the actual content of our work, we are worried that there is little chance for a fruitful discussion here. We have flagged the review to the ACs as a potential conflict of interest, and we refrain from commenting further.

---

> > ### Comment · Reviewer_gz9E · 2023-11-22
> >
> > My review has a confidence of 5 because the topic of this paper is precisely my area of expertise, and because I have worked through both D-adaptation and the paper under review in detail. I gave the lowest scores because the paper contains several inaccuracies, multiple theorems with incorrect proofs (the lower bounds), and fails to make meaningful improvements over the existing works. I am not sure how any of that implies a conflict of interest; it seems to me that the authors are trying to exclude actual experts in this topic while making ambiguous claims about the novelty of their results.
> >
> > To re-iterate: the theory fails to achieve any new results that haven't already been achieved in much harder problem settings in the online learning literature. Thus there should at the very least be some fair empirical demonstration showing that Prodigy has some advantage in practice over the existing methods, yet not one of the many existing works that already achieve stronger results than Prodigy has been included as a baseline --- for all we know Prodigy could have worse theoretical guarantees *and* worse practical performance than existing methods.
> > For a detailed discussion of these issues, please see my review.

---

### Official Review · Reviewer_bqiE · 2023-11-01

**Soundness:** 3 good
**Presentation:** 3 good
**Contribution:** 3 good
**Rating:** 6
**Confidence:** 4

**Summary:**

The authors propose two different improvements to the award-winning learning-rate-free D-adaptation algorithm, termed Prodigy (product of D and G), in turn, with GD, Dual Averaging and Adam versions, and D-adaptation with Resetting.
Both approaches improve the non-asymptotic bound for D-adaptation by removing a log factor, but seem to add other factors of their own.
The authors also prove a technical result that among exponentially bounded algorithms, a new characterization of algorithms, their D-adaptation variants are optimal.
In a series of convincing experiments, similar to the ones in the original D-adaptation paper, the Adam Prodigy variant is found to  be comparable (possibly slightly better in some cases) to the D-adaptation Adam variant.
The resetting approach is not experimentally evaluated as it isn't expected to outperform Prodigy.

**Strengths:**

1. The paper is well-motivated and appears to be theoretically strong. (This reviewer didn't check the proofs though.)
2. The experimental results confirm the theoretical guarantees for the convex logistic loss.
They also show that Prodigy and D-adaptation perform similarly, possibly slightly better in some cases, on small and large neural networks with non-convex losses despite the lack of theoretical guarantees.
3. Experimental results in Fig. 1 seem to show the apparently new result that D-adaptation as well as Prodigy outperform the recently proposed DoG and L-DoG algorithms.

**Weaknesses:**

1. Both approaches appear to be more complex than the original D-adaptation approach, by introducing additional weights,
and the practical benefit of the newer theoretical guarantees is not clear.
As mentioned, they remove one factor from the non-asymptotic bound of D-adaptation, but replace it with another.
It is not clear how much tuning was needed to obtain the small occasional improvements over D-adaptation in Figures 1-3.

2. The paper is also marred by symbolic confusion and occasional grammatical errors, e.g.,

    2a. line 4 in Algorithm 2 mentions $\lambda_k = d_k^2$, but the fifth line of Sec. 2 mentions $\lambda_k = d_k$

    2b. Grammatical error or typo in Sec 5, page 7: "...can divergence in theory..."

    2c. Grammatical error or typo in Sec 5, page 7: "...initial sub-optimally of the D estimate ... "

**Questions:**

Please note the inherent question underlying weakness 1.
It is mentioned below Theorem 1 regarding the careful setting of $\lambda_k$, "While it is not guaranteed to be better in theory, it is usually quite important to do so in practice." However, Sec 6 shows no great practical benefit from the apparent careful setting of $\lambda_k$.
There is apparently some slight benefit in test losses for ViT for Resnet-50, but it is not clear how much tuning of $\lambda_k$ was done to achieve this benefit.
The main promise of D-adaptation is that one does not need to perform much tuning of such optimization parameters.

---

> ### Author Response · Authors · 2023-11-22
>
> Thank you for providing us with feedback on our work. We address the weaknesses and answer your question below:
> W1. As you correctly pointed out, our dual averaging method does not have the $\log(n)$ factor, but this is actually not the main contribution of our work (D-Adaptation also has a dual-averaging version). The most important part is that our methods have guarantees proportional to $\sqrt{\log(D/d_0)}$, whereas D-Adaptation-based methods have guarantees with factor $\log(D/d_0)$, the latter of which is always larger.
> W2. Thanks for bringing the notation and grammar issues to our attention, we will change the notation to make it clearer, especially for  $\lambda_k$, and we'll do an extra pass to fix the typos.
> Q. In all of our experiments, we simply set $\lambda_k=\beta_2^{-k/2}$, which is derived in Appendix B as the right way to get exponential moving averages of the gradient magnitudes. Therefore, Prodigy did not need any tuning. You can also see that Algorithm 4, which we used for the experiments, doesn't require $\lambda_k$ as input. The main reason we introduced $\lambda_k$ in the analysis was to 1) justify EMA and 2) make sure that $d_k$ is still estimated the right way. Furthermore, we didn't tune $\beta_2$ and used the value 0.999 for all methods.
> As the reviewer pointed out, the main benefit comes on vision tasks when training ViT on ImageNet and VGG/ResNet on CIFAR10, giving 1-2% improvement in test accuracy. We want to emphasize that this came at no extra cost and we didn't do any special tuning to achieve the improvement.

---

> > ### Comment · Reviewer_bqiE · 2023-11-23
> > **Thank you for the clarifications !**
> >
> > The authors have adequately addressed my concerns and answered my question satisfactorily.
> > However, given the comments from an expert reviewer who has more concerns about this paper, I've realized that I should have been less confident in my original positive review. While I will retain my positive review, I would like to lower my confidence from 4 to 3.

---

### Official Review · Reviewer_UUCM · 2023-11-01

**Soundness:** 3 good
**Presentation:** 2 fair
**Contribution:** 2 fair
**Rating:** 5
**Confidence:** 3

**Summary:**

In this submission, the authors improve on D-adaptation---a method for deterministic non-smooth convex optimization with promising performance in practice---by changing the step-sizes to be normalized not by the sum of the square-root of the norm squared of the gradients seen so far, but by the product of these norms squared with the current (lower) estimates of the distance of the initial point to the optimum. They also add extra parameters to allow for "step-size schedules", which do not affect the theoretical results but are important for the empirical performance. They show improved theoretical convergence rates (shaving off a square root log factor), and also show that a restarted version of D-adaptation also enjoy similar guarantees. Moreover, the authors also show a few lower-bounds for convex non-smooth optimization. Finally, they conclude by showing consistent improvement of Prodigy over D-adaptation over a large array of deep learning optimization tasks.

**Strengths:**

The methods described do seem to improve over D-adaptation, either by changing the denominator in the step-sizes used or via resetting. The fact that these improve on the convergence guarantees from D-adaptation and show consistent improved performance on a variety of deep learning tasks is interesting.

**Weaknesses:**

The main weakness of the paper might boil down to presentation, but I am having a hard time correctly understanding many of the contributions. I am mostly knowledgeable on the theoretical aspects of optimization and online learning (although I am not closely acquainted with parameter-free methods in online learning). Maybe some of my worries are due to a lack of background from my part, At the same time, I consider myself a researcher with more knowledge in these fields than the average person in the community. So I do think these worries would be shared by many people that even work in topics related to this paper. I hope my questions and discussion with authors and reviewers help me arrive to a fair assessment of the paper. Of course, feel free to let me know if I am missing a major point of the paper or a big piece of the literature that was not mentioned in the prodigy paper due to space constraints. The main point is likely the first one, the other ones are minor points.

**Prodigy convergence rates vs rates in more general settings**: Theorem 1 and 2 show that prodigy improved on the learning rate of D-adaptation by a factor of $\sqrt{\log(D/d_0)}$. It is not clear if this improvement should be expected to be reflected in empirical problems since in practice we see convergence rates that are much faster than $1/\sqrt{t}$. In this case, if the rate does not reflect what happens empirically, I would guess that getting better rates is important for its novelty among other convergence rates. However, in the related work section the authors say that this new rate matches the one from online learning, which is a much more adversarial setting. In the stochastic case, which more closely matches the deep learning optimization case, the work of Carmon and Hinder (2022) have a $\log \log (D)$ factor in the convergence rate for the deterministic version of the algorithm if I am not mistaken, which is better than Prodigy's convergence rates. However, this is mostly a theoretical work, and the goal of Prodigy is to also be practical. On this line, the authors claim that they improve on the DoG convergence rates by shaving off a $\sqrt{\log D/d_0}$ factor. The DoG rates are for the stochastic setting with locally Lipschitz gradients, significantly more general than the deterministic setting where the rates of Prodigy hold.

So in the end I am not able to grasp what is the relevance of the improved theoretical convergence rates. It is in fact interesting that the improvement in the empirical performance might be connected to this slightly tighter convergence rate. But it is not clear how connected they are, if at all, and the paper does not discuss this connection. In the purely theoretical side, the rates at match (or slightly improve in the case of DoG) over other convergence rates, but on a more restricted setting.


**Comparison with other algorithms**: Although the empirical results are definitely the strongest part of the paper, I did not understand why the only comparison point in the experiments is D-adaptation and DoG. Maybe this is discussed in a part of the appendix that I have not read (if so, please let me know), but is it the case that any other algorithms perform poorly enough that they are not worth considering in these comparisons?

---

Here are a few minor suggestions:

**Lower bounds and the dependency of $n$ and $D$**: I thought the lower bounds were interesting, but the fact that the function $f$ (and, thus, $D$) depend on $n$ in such a way that $\log \log D/d_0$ is roughly of order $\sqrt{n}$ seems very important and a big reason why these lower bounds do not rule out the possibility of improved asymptotic convergence rates, even in the stochastic case. Although this is mentioned by the authors before the lower bounds, adding that this is the case in the theorem statements themselves would make them more readily understandable by people. Mentioning that the function is of the form $f(x) = |x - x^*|$ could also be interesting, if space allows.



**Typos and crammed equations**: As a last minor point, there are several equations that are crammed in the paper (sometimes with parts going on top of the text). I understand the submission process can be rushed, but if the authors could do a revision pass of the paper, it would be great.

**Questions:**

So here are my main question that I would appreciate if the authors could comment on.

- Do you have some explanation of why/how the improved convergence rates should impact practical performance of the algorithm? In practice we see a convergence rate that is much faster than a $O(1/\sqrt{n})$ rate, and at this point it is not clear why shaving off the $\sqrt{\log D / d_0}$ factor matters, even more so considering that these rates are only proven for the deterministic setting, while the empirical performance is studied in the stochastic case.
- Could the authors expand on the relevance of the convergence rates given the comparisons I described with other works in more general settings? A big focus of the paper is put on the convergence rates, but the rates by themselves do not seem relevant if compared to other works, so I am probably misunderstanding something. Could the authors clarify this point? At this stage, it feels like this is a interesting contribution for deep learning optimization, but most of the paper was written trying to frame the contribution as a theoretical one.

---

> ### Author Response · Authors · 2023-11-21
>
> Thank you for your detailed review! We found your feedback to be very insightful and we we hope we can clarify some of the confusing parts:
> 1. Our main contribution is modifying D-Adaptation to adapt the learning rate faster. D-Adaptations at its core is a method that builds directly on top of Adam by estimating the learning rate, with the theory based on Adagrad. While the reviewer is right that the theoretical rate is slower than what we see in practice, D-Adaptation's main contribution is the theory-based estimate of the learning rate, and we are improving this specific aspect of the theory. The fact that D-Adaptation works well in practice suggests that this aspect of the theory is useful, so our $\sqrt{\log(D/d_0)}$ improvement might translate to a better practical performance. As we then validated in the paper using an extensive suite of benchmarks, there is indeed an improvement in some cases (ImageNet, CIFAR-10) and in other cases it works the same way as D-Adaptation and hand-tuned Adam.
> 2. Indeed, online learning and the stochastic setting of DoG are more general, which we will clarify in the updated version of the paper. We again highlight that our main point is improving upon D-Adaptation, which is of interest due to its practical performance.
> 3. Adam with a hand-tuned learning rate has been the main baseline in the last years and we are not aware of any optimizer to consistently outperform it. Adam was also the main baseline in the DoG paper, while the D-Adaptation papers included as a baseline coin betting, which did not perform well. If there is any specific optimizer you believe works well, we'll be happy to try it.
> 4. We agree with your comments on the lower bounds.
> 5. The ICLR page limit turned out to be very tight, so we squeezed the text a bit too much, we will fix the formatting issues.
>
> To conclude, we thank the reviewer for pointing out the confusing parts. We do believe that the convergence rates are important, but ultimately what works best is often determined by other factors that are not directly connected to the rates. We aim to clarify this aspect of our work in the revised version.

---

> > ### Comment · Reviewer_UUCM · 2023-11-22
> > **Thanks for the response and a few comments**
> >
> > I'd like to thank the authors for taking the time to address some of the concerns raised. I think the first comment is really the main reason we might have seen somewhat negative reviews, although this does not mean I agree with the other reviews and less so with the way the points were raised.
> >
> > More specifically, the authors say:
> > > While the reviewer is right that the theoretical rate is slower than what we see in practice, D-Adaptation's main contribution is the theory-based estimate of the learning rate, and we are improving this specific aspect of the theory. The fact that D-Adaptation works well in practice suggests that this aspect of the theory is useful, so our improvement might translate to a better practical performance.
> >
> > I think the issue is that this claim is almost self-contradicting (emphasis on *almost*). The authors say that the main contribution of D-adaptation is the *theory-based* estimate of the learning rate. However, the theory does not reflect what happens in practice directly, *although the original D-adaptation paper and this one makes it seem that there is a connection*. I do agree that this is super interesting, but I disagree when the authors frame the paper's main contribution as being the "theory-based estimates of the learning rate", mainly because the connection with the theory is not yet clear, theoretically this results are weaker than other results, and if it were not for the great performance of D-adaptation in practice, I believe the theoretical contributions would not be interesting.
> >
> > I have a suggestion, but take this as it is: a suggestion from a conference reviewer with limited time, so take them with a grain of salt. Moreover, I do not want to be arrogant, so this really should be taken as a suggestion, not as something that I am completely certain of. With these qualification, *I believe the paper should be framed as a practical improvement to D-adaptation*, and then mention that the theory also is improved if compared to D-adaptation, and possibly these 2 things being connected is actually an interesting observation. But the focus of the paper on claiming the theoretical aspects of the method as the main contribution seems, to me, to be the big detractor in most of the reviews.
> >
> > It seems to me that Prodigy is of interest to the community, but I'd still vouch for rejection of the paper because of the problems on discussion with related work but, mostly, because it seems to put too much emphasis on the theoretical aspects, when in fact the main contribution is a mix, leaning more towards to practical side: it is a method that has great practical performance and *it seems like there is a connection with the theory of it*, but it is hard to know at this point if this is an actual connection to theory of just spurious correlation.

---

> > > ### Author Response · Authors · 2023-11-22
> > >
> > > We appreciate your candid feedback, and thank you for the advice!
> > > Your suggestion to frame the paper as a practical improvement to D-Adaptation makes sense from the perspective of Prodigy being useful in practice. Our perspective is that it's not the end yet as many components are not yet understood, including momentum, weight decay, and schedulers. We believe the way forward is to develop the theory further, which is why we emphasized it in the paper. For instance, we are working on estimating $\beta_1$ and $\beta_2$ adaptively, and it is all based on the theory rather than the empirical improvements. The main value of the empirical improvements, in our opinion, is to verify that the developed theory is a step in the right direction.
> > > Your perspective is nevertheless very helpful to us to see how the paper can be improved and we'll definitely take it into account.

---

### Official Review · Reviewer_JC31 · 2023-11-06

**Soundness:** 3 good
**Presentation:** 2 fair
**Contribution:** 2 fair
**Rating:** 5
**Confidence:** 2

**Summary:**

This paper provides a modification of D-Adaptation to improve its worst-case non-asymptotic convergence rate for a G-Lipschitz objective. D-Adaptation’s convergence rate scales with $\frac{\log(D/d_0)}{\sqrt{n}}$. Prodigy (the paper’s modification of D-Adaptation) has a convergence rate that scales instead with $\frac{\log(n)\sqrt{\log(D/d_0)}}{\sqrt{n}}$. Asymptotically, Prodigy is slower due to the additional $\log(n)$ in the numerator, but faster in finite time due to the scaling of $\log(D/d_0)$.

The main point is that Prodigy effectively uses larger step sizes. Prodigy replaces the D-Adaptation learning rate $\frac{d_k}{\sqrt{G^2 + \sum_{i=0}^k \|g_i\|^2}}$ with $\frac{d_k}{\sqrt{\frac{1}{\lambda_k^2} G^2 + \sum_{i=0}^k \left[\left(\frac{d_i \lambda_i}{d_k \lambda_k}\right)^2 \|{g_i}\|^2\right]}}$. The estimates of D, $d_i$, are non-decreasing, so for the right choices of $\lambda_i$, the learning rate is effectively larger.

They show improved training losses from D-Adaptation for deep learning tasks on various architectures, sometimes doing better than Adam.

**Strengths:**

- Empirically, there are some observable improvements with Prodigy versus D-Adaptation.
- The derivation/algorithm seems to be sound.
- I am not an expert in this area, so leave no comments about the novelty.

**Weaknesses:**

I’m mostly concerned about some of the claims made in the non-convex deep learning section, beyond just the fact that what any of these algorithms are really doing deep learning is unclear.

- In terms of practical impact, the improvements made by Prodigy reported Figure 1/2/3 are small. Often the differences between Adam, Prodigy, and D-Adaptation are much smaller than their standard errors. It doesn’t always do better either, superseded by Adam, and sometimes D-Adaptation.
- Adam’s initial learning rate should be hyperparameter tuned.
- They start to make claims about the test accuracy / generalization that I don’t think they should/need to include. Specifically, they make this claim about CIFAR10 on ResNet and ImageNet on Vision Transformers. In both scenarios, the models are overfitting, trained for hundreds of epochs on the training data. Lower training loss here does not necessarily mean higher test accuracy.
    - Under Figure 1, they write “Prodigy estimates a larger step size than D-Adaptation, which helps it reach test accuracy closer to the one of Adam.” (They aren’t talking about  any implicit bias of large learning rate, just strictly that large learning rate leads to better training loss convergence.)
    - Under paragraph “ViT training” they write “Prodigy almost closes the gap [of test performance] between tuned Adam and D-Adaptation.”

Some other minor comments:
- Can Figure 2 be plotted in log scale? It’s hard to see the difference between the lines.
- Text spacing issue in Page 3

**Questions:**

See weaknesses

---

> ### Author Response · Authors · 2023-11-21
>
> We thank the reviewer for their time and effort to give us feedback. We can see that some of our claims were confusing, so we would like to clarify some of them:
> 1. We believe that the improvements made by Prodigy on Imagenet and CIFAR-10 are significant enough (roughly 2% in test accuracy) to suggest the use of Prodigy over D-Adaptation. Moreover, in cases where the improvements are small, in particular when D-Adaptation already matches the hand-tuned Adam, there is no downside to using Prodigy.
> 2. We did tune the initial learning rate of Adam, otherwise we agree it wouldn't have been an interesting comparison.
> 3. The claims we made about test accuracy are only coming from the observed values. Figures 1, 2, 3 all report the test accuracy/perplexity in the left column as the main metric. We did not intend to make any claims about the generalization properties of Prodigy beyond stating that we observe better values and that we observe larger learning rates, we will make it clear that we are not claiming a better implicit bias from the larger learning rate. We also highlight that we used weight decay and standard data augmentations, which is why the test accuracy keeps improving even after 100 epochs.
> 4. The point of Figure 2 is mostly that there is no difference between the lines, it illustrates that we get the same performance without tuning the learning rate. In fact, Figure 2 looks almost the same way in log scale (the curves' shapes remain the same) since the values are all above 1 and do not go to 0.
> 5. Thank you for pointing this out, we will fix the formatting issues on page 3.

---

### Meta-Review · Area_Chair_z14L · 2023-12-21

**Metareview:**

After careful examination of the reviews and the authors' responses, the consensus is to reject the submission. While the paper presents modifications to the D-Adaptation algorithm and claims both theoretical and empirical improvements, the criticisms raised by Reviewer gz9E, in particular, highlight substantial issues that have not been satisfactorily addressed by the authors.

Reviewer gz9E, who demonstrates expertise in the area of online learning, points out that the theoretical contributions are not novel and that the improvements claimed over D-Adaptation have already been achieved in more general and challenging settings. The reviewer also raises valid concerns about the validity of the lower bounds provided in the paper and the lack of comparison with established baselines from the relevant literature.

Although Reviewers UUCM and JC31 acknowledge the effort to improve upon D-Adaptation and the potential practical benefits, they too express reservations about the significance and clarity of the theoretical contributions. Reviewer UUCM suggests that the paper may be over-emphasizing the theoretical aspects when the main value lies in practical improvements. Reviewer JC31 questions the degree to which the algorithm truly avoids the need for tuning hyperparameters, which is a central promise of the original D-Adaptation algorithm.

Reviewer bqiE initially provided a more positive assessment, highlighting the paper's motivation, theoretical strength, and confirmation of theoretical guarantees in convex scenarios. However, upon reflection and in light of the other reviews, this reviewer has revised their confidence level downwards.

In response to the reviews, the authors argue that their main contribution lies in the theory-based estimate of the learning rate and that their empirical improvements validate the theory's relevance. However, the connection between the theoretical improvements and practical performance is not convincingly established in the paper.

Given the substantial concerns about the paper's novelty, theoretical soundness, and empirical validation, the committee recommends rejection. The authors are encouraged to address the raised issues and consider re-framing their contributions in future submissions.

**Justification For Why Not Higher Score:**

A higher score is not justified due to the lack of novelty in the theoretical contributions, incorrect lower bounds, and insufficient empirical validation against relevant baselines from the online learning literature. The concerns raised by Reviewer gz9E, who is an expert in the field, are particularly persuasive and have not been effectively countered by the authors.

**Justification For Why Not Lower Score:**

N/A

---

### Decision · Program_Chairs · 2024-01-16

Reject